# From TDP-43/RNA complex formation to disease-linked TDP-43 aggregation through a structural and cellular approach

Yitian Feng[1], Vandana Joshi[1,3], Serhii Pankivskyi[1,3], Marie-Jeanne Clément[1,3], Juan Carlos Rengifo-Gonzalez[1], Aurélien Thureau [2], David Pastré [1] & Ahmed Bouhss [1]

Many RNA-binding proteins (RBP) have been associated to several neurodegenerative diseases for which RBP-rich cytoplasmic inclusions represent a major histological hallmark. However, among RBPs, the occurrence with which TDP-43, a nuclear mRNA-binding protein, is detected in cytoplasmic inclusions is exceptionally high. To unravel the underlying mechanisms, we focus our analysis on the structured N-terminal domain (NTD) of TDP-43, which is distinct among RBPs as this domain mostly initiates TDP-43 homotypic interactions. Through an in depth structural analysis, we successively show that the cooperative binding of TDP-43 along long GU-rich intronic sequences antagonizes NTD/NTD interactions between adjacent TDP-43 along mRNA. In contrast, the TDP-43 cooperativity facilitates NTD/NTD interactions between TDP-43 located on distinct GU-rich sequences. We hypothesize that NTD/NTD interactions between distinct GU-rich sequences efficiently allow the compaction of long introns in neurons under physiological conditions. However, when the binding of TDP-43 to RNA is discontinuous because of a lack of cooperativity, aberrant NTD/NTD interactions between adjacent TDP-43 take place, promoting the aggregation of TDP-43 RRMs (RNA Recognition Motifs) under stress conditions. Altogether, we provide a detailed view of the physiological assembly of TDP-43 on introns and the putative weaknesses of TDP-43 that makes it distinct in its propensity for aggregation compared to other RBPs.

Many RNA-binding proteins (RBP) in the human genome harbor low complexity domains that initiate weak but numerous interactions, leading to the formation of subcellular compartments formed in the absence of membranes[1]. Under physiological conditions, RNA-related membraneless compartments that are mostly located in the nucleus, such as paraspeckles, speckles and nucleoli, fulfill many complex functions related to RNA transcription, mRNA splicing and ribosomal RNA processing[2–4]. However, in several neurodegenerative diseases such as amyotrophic lateral sclerosis (ALS) and frontotemporal lobar degeneration (FTLD), nuclear RNA-binding proteins abnormally form cytoplasmic inclusions. Nuclear RBP aggregation in the cytoplasm potentially induces a loss of their nuclear function and/or a gain of toxic function in neurons[5–8]. Among the nuclear RBPs mislocalized in the cytoplasm, TDP-43 (encoded by TAR DNA Binding Protein, *TARDBP*

[1]Université Paris-Saclay, INSERM U1204, Univ Evry Paris-Saclay, Structure-Activité des Biomolécules Normales et Pathologiques (SABNP), Evry-Courcouronnes, France. [2]Synchrotron SOLEIL, L'Orme des Merisiers, Départementale 128, Saint Aubin, France. [3]These authors contributed equally: Vandana Joshi, Serhii Pankivskyi, Marie-Jeanne Clément. ✉e-mail: david.pastre@univ-evry.fr; ahmed.bouhss@univ-evry.fr

gene) is the most frequently detected protein in cytoplasmic inclusions of ALS- and FTLD-affected neurons[9–12]. Since many of the TDP-43 pathological mutations are located in the C-terminal low complexity domain[13] which is known for promoting phase separation, a transition from reversible TDP-43 liquid-like assemblies to an irreversible solid-like TDP-43 aggregation[14–16] has been considered as the main factor of TDP-43 aggregation. However, pathological mutations in low

complexity domains are not restricted to TDP-43 since they can be found in other RNA-binding proteins such as FUS[17]. The specificity of TDP-43 rather relies on its structured N-terminal domain (NTD) which is located at the N-terminal end (a.a. 4-77, Fig. 1A). TDP-43 NTD is known for its capacity to oligomerize[18,19]. Interestingly, besides NTD auto-assembly, no other protein has been identified as interacting directly with NTD or competing for the NTD-mediated

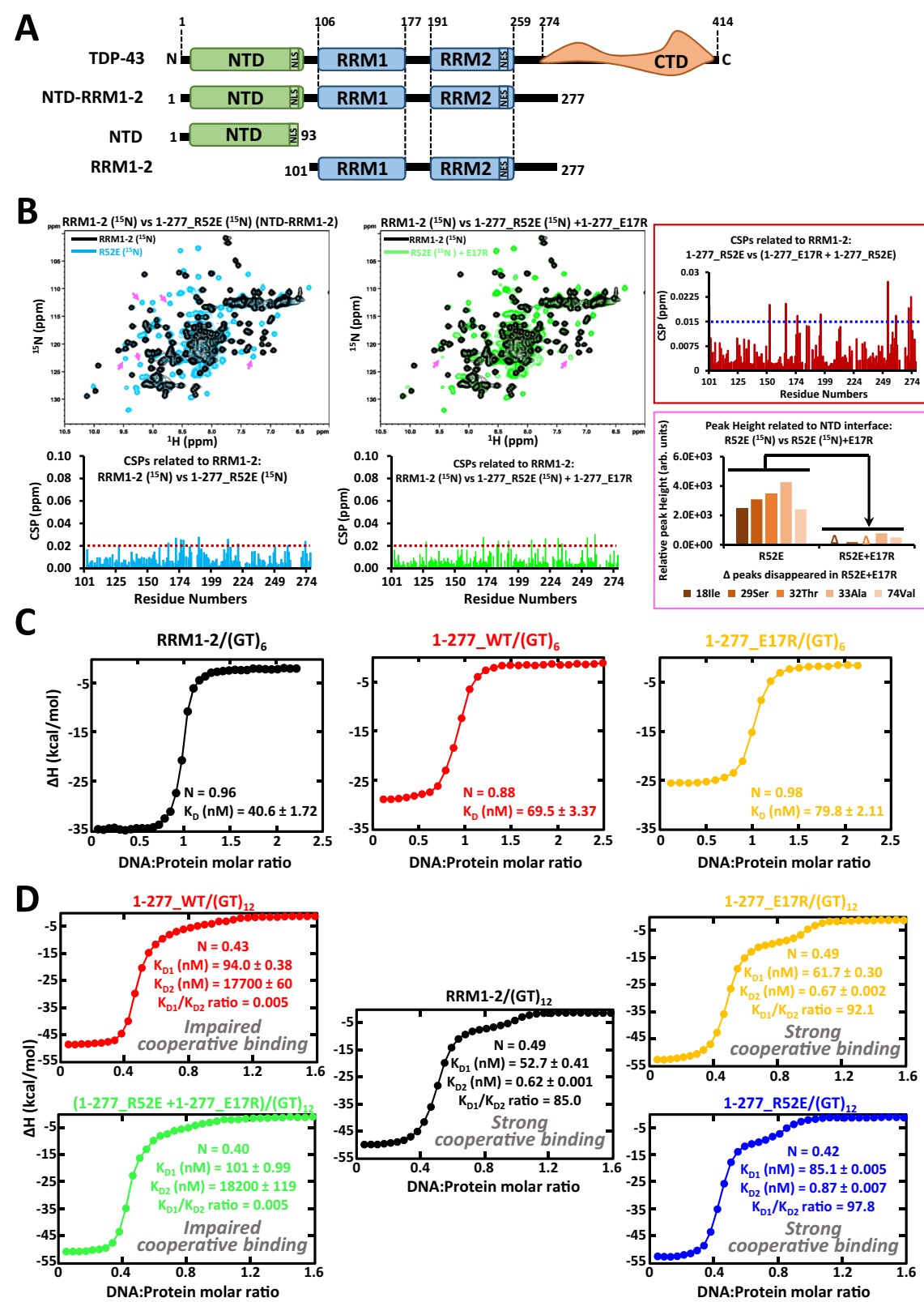

**Fig. 1 | NTD/NTD interactions are not compatible with the cooperative binding of TDP-43 along RNA. A** TDP-43 constructs used in the in vitro assays. **B** Upper left panels: Superimposed NMR spectra of [15]N-labelled RRM1-2 (a.a., 101-277) and mutated TDP-43 (a.a., 1-277) under indicated conditions. Lower left panels: Chemical Shift Perturbations (CSPs) of indicated residues when comparing RRM1-2 residues. Red dotted line shows three times the standard deviation of the CSP values. Residues without a bar were not assigned or overlapped in 2D spectra. Upper right panel: CSPs of RRM1-2 residues after reconstituting NTD/NTD interactions by adding unlabeled E17R (a.a., 1-277). Blue dotted line shows three times the standard deviation of the CSP values. Lower right panel: Peak heights of NTD residues with and without NTD/NTD interactions. NMR peaks of these residues are indicated by pink arrows in 2D spectra. **C** Left panels: ITC titration curves and affinity parameters for (GT)$_6$ oligonucleotides that can bind on one protein monomer. **D** Upper panels: ITC titration curves and affinity parameters for indicate constructs or equimolar mixture of indicated constructs interacting with (GT)$_{12}$.

multimerization. In addition, to our knowledge, no RBP other than TDP-43 itself harbors a similar NTD domain. The multimerization of TDP-43 NTD has been the subject of many studies to understand its potential role in the translocation of TDP-43 to the cytoplasm[20,21], in TDP-43 aggregation[20,22], and in its physiological role in splicing process[18,23]. Several reports indicated that NTD intermolecular interactions promote aberrant phase separation[18,24]. Especially, the formation of head-to-tail linear chains of NTD domain[18] is considered as a major intermediate leading to TDP-43 aggregation in several models[20]. However, there is no structural data on the higher order assembly of TDP-43 when TDP-43 is complexed with long intronic GU-rich RNA sequences, its favorite nuclear RNA targets. The precise structural contribution of NTD/NTD interactions to the physiological role of TDP-43 in the processing of introns, as well as the interplay between TDP-43:RNA and NTD/NTD interactions in the structural transition from reversible RNA-rich TDP-43 assemblies to irreversible RNA-free TDP-43 aggregates, still remain to be dissected. Addressing these points may be critical to understand the successive steps leading to TDP-43 aggregation in neurodegenerative diseases.

Here, to tackle the above-mentioned issues, we first considered the higher order assembly of TDP-43 in the presence of GU-rich RNA. TDP-43 RRM1-2 binds cooperatively to GU-rich RNA based on an intermolecular interaction between RRM1 and RRM2[25]. The cooperative binding of TDP-43 along RNA may either promote or prevent NTD/NTD interactions between adjacent TDP-43. Therefore, to document the interplay between the cooperative binding of TDP-43 to RNA and NTD/NTD interactions, we undertook a comprehensive structural characterization based on the analysis of extensive biochemical and NMR data using a long fragment of TDP-43 that contains both the NTD and the two RRMs. The structural analysis was complemented by thermodynamic data based on ITC analyses and large-scale views on the higher order assemblies of TDP-43 in complex with long oligonucleotides provided by Small Angle X-Ray Scattering (SAXS). We concluded that TDP-43 assemblies along GU repeats antagonize intermolecular NTD/NTD interactions between adjacent TDP-43 along RNA. On the other hand, long-range interactions between distant TDP-43:GU-rich RNA clusters can induce the compaction of long mRNAs. Accordingly, we then demonstrated that NTD-mediated interactions as well as the cooperative binding of TDP-43 to mRNA contribute to the formation of TDP-43 higher order assemblies in cells. Their contributions are interrelated since TDP-43 cooperative associations along GU-rich sequences favor long-range NTD/NTD interactions between distinct clusters and vice versa. To investigate TDP-43 aggregation, cells were exposed to mild oxidative stress in the presence of arsenite at low concentration. We evidenced a massive TDP-43 aggregation promoted by NTD/NTD interactions only when the cooperative association of TDP-43 to GU-rich mRNA was impaired. Therefore, in the absence of cooperative association, aberrant NTD/NTD interactions among TDP-43 adjacently bound to RNA promote TDP-43 aggregation. TDP-43 aggregation is due to inter-RRM irreversible assembly linked to the acetylation of some RRM lysine residues under oxidative stress conditions, as previously reported[26]. Interestingly, in agreement with the presented model, the expression of introns harboring long GU repeats in the nucleus significantly reduces the aggregation of wild type TDP-43 in the nucleus after arsenite stress.

Altogether, the combination of biochemical, structural and cellular data presented here sheds light on the aggregation process of TDP-43 in major neurodegenerative diseases such as ALS and FTLD. Under physiological conditions, the cooperative association of TDP-43 to GU-rich RNA and NTD linked to interactions provides an efficient mean to compact long TDP-43 targeted introns, rich in GU stretches. However, in the cytoplasm, the scarcity of GU-rich sequences in mature mRNA may limit the cooperative association of TDP-43. Therefore, discontinuous associations of TDP-43 along cytoplasmic RNA promote aberrant NTD/NTD interactions to facilitate TDP-43 cytoplasmic aggregation. We anticipate that a better understanding of the multiple steps leading to TDP-43 aggregation would help to design strategies aiming at limiting TDP-43 aggregation to prevent the onset or dampen the progression of TDP-43-associated neurodegenerative diseases.

## Results

### NTD/NTD interactions antagonize TDP-43/GU-rich cooperativity

In previous investigations at atomic level, we demonstrated that the TDP-43 tandem RRMs, RRM1-2 (a.a.,101-277), bind cooperatively to GU-rich RNA or GT-rich single-strand DNA repeats[25]. The cooperative binding occurs through an intermolecular interaction between the pocket around V220 in RRM2 of one monomer and the RRM1 loop 3 of an adjacent TDP-43 monomer when TDP-43 binds to GU-rich RNA or GT-rich ssDNA[25]. We then questioned the possible role of the structured NTD domain in the cooperative association of TDP-43 to RNA. At first sight, a link between two monomers via their respective NTDs may promote or secure the cooperative association (Hypothesis 1). However, conversely, the spatial arrangement resulting from the cooperative association may cause NTDs to be well separated from each other along the GU repeats. Therefore, the interaction between NTDs could be incompatible with a cooperative binding of TDP-43 to GU and GT repeats. In this case, NTD/NTD interactions may provide a negative contribution to the cooperative binding of TDP-43 to RNA or ssDNA (Hypothesis 2).

To challenge both hypotheses, we used a longer TDP-43 fragment (a.a., 1-277), including the full NTD domain, sufficiently soluble to be studied in vitro by using structural biology tools (ITC, NMR spectroscopy) (Fig. 1A, Supplementary Fig. 1 and 2A). We first verified that the presence of the NTD domain does not induce major structural changes or new interactions with RRM1-2. For this, we compared the chemical shifts of the RRM1-2 residues when the fragment contains or lacks the NTD domain (a.a., 1-277 *vs* a.a., 101-277, Fig. 1B and Supplementary Fig. 2A, B). We also constructed two NTD mutants, R52E and E17R, which significantly impair the NTD/NTD multimerization on opposite sides of the NTD domain and display the molecular weight of a protein monomer (Supplementary Fig. 1). In the absence of RNA or ssDNA, the peaks of RRM1-2 residues are superimposable for all the fragments tested (NMR spectra in Fig. 1B and Supplementary Fig. 2B). In addition, we reconstituted the NTD/NTD interface with an equimolar protein mixture of [15]N-labelled R52E and unlabeled E17R (R52E + E17R, a.a., 1-277). The reconstitution of NTD/NTD interactions involves only the non-mutated face of each complementary mutants E17R and R52E. In this case, the peaks of NTD residues involved in NTD/NTD multimerization disappear (as compared to respective residues in NTD

mutant alone), attesting to the recovery of the NTD/NTD interface (Fig. 1B). Each mutant contributes to form the dimer interface through its non-mutated side as confirmed from NMR experiments by using [15]N-labelled R52E or [15]N-labelled E17R (Supplementary Fig. 3). However, no residue or group of residues displays major chemical shifts in RRM1-2 due to NTD/NTD interactions (Fig. 1B and Supplementary Fig. 2B).

Next, we probed whether the cooperative association of TDP-43 to GT repeats depends on NTD/NTD interactions by using ITC analysis. We first tested a stretch of 6 GT repeats, $(GT)_6$, which can only accept one TDP-43 unit. In this case, the presence of the NTD, whether functional (WT) or not (NTD mutant, E17R), does not affect the stoichiometry nor the affinity of TDP-43 for $(GT)_6$ (Fig. 1C, Supplementary Fig. 4A).

In the presence of $(GT)_{12}$ (12 GT repeats), two RRM1-2 (a.a., 101–277) can bind to the same oligonucleotide (Fig. 1D and Supplementary Fig. 4B; RNA/protein molar ratio, N ~ 0.5), which leads to the presence of 2 plateaus, a major feature of cooperative association to $(GT)_{12}$. The calculations confirmed the strong cooperativity of isolated RRM1-2 alone since the value of the ratio, $K_{D1}/K_{D2}$ is very high, meaning that the binding affinity of the second monomer to $(GT)_{12}$ is stronger than the first one. However, when we used the long fragment which contains both RRM1-2 and NTD (a.a., 1-277), the cooperative phenomenon is strongly disrupted (low $K_{D1}/K_{D2}$) suggesting that NTD/NTD interactions are an obstacle for the cooperativity (red plot in Fig. 1D). To confirm this point, we also performed ITC experiments by using the R52E mutant, which prevents the intermolecular association between NTDs. The ITC profile shows that the cooperative association of TDP-43 R52E mutant to $(GT)_{12}$ oligonucleotide is then restored (blue plot in Fig. 1D). TDP-43 E17R mutant displays similar behavior as R52E mutant (yellow plot in Fig. 1D). Importantly, when we generated NTD/NTD interactions with an equimolar mixture of R52E and E17R mutants (reconstituted dimer, R52E + E17R, a.a., 1-277), the cooperative assembly of TDP-43 to $(GT)_{12}$ once again significantly decreases (green plot in Fig. 1D, Supplementary Fig. 4B). Moreover, TDP-43 targets specifically GT/GU repeats. Accordingly, TDP-43 displays no affinity for poly (A) sequence as showed by using EMSA (Supplementary Fig. 5A) and very limited chemical shifts in the presence of poly (A) sequence or CA repeats in NMR spectra (Supplementary Fig. 5B). In this case, the introduction of a stretch of poly-A $(A_{12})$ between two $(GT)_6$ repeats, interrupting the possibility of continuous binding of R52E to GT-rich ssDNA, prevents the cooperative association of R52E (Supplementary Fig. 4C). Together, the results clearly confirm the negative contribution of NTD/NTD interactions to the cooperative binding of TDP-43 to long $(GT)_{12}$ repeats.

## NTD does not interfere with RRM1-2 binding to nucleic acids

To further document the interplay between the cooperative binding to nucleic acids and NTD/NTD interactions, we performed an extensive NMR analysis of wild type TDP-43 and R52E in the presence of 6 or 12 GU/GT repeats. By comparing the residues involved in RNA interactions in the wild type RRM1-2 (a.a., 101-277), wild type TDP-43 (a.a., 1-277), and TDP-43 R52E mutant (a.a., 1-277), we found that NTD/NTD interactions did not significantly interfere with the chemical shifts of RNA-binding residues in RRM1-2 domains in the presence of short $(GT)_6$ oligonucleotide (Supplementary Fig. 6 and 7). We can therefore conclude that NTD/NTD interactions have no impact on the binding of RRM1-2 to short nucleic acids. To confirm this point, we also used an equimolar mixture of [15]N-labelled R52E and unlabeled E17R mutants (a.a., 1-277) to reconstitute a R52E/E17R dimer, as demonstrated by the disappearance of the resonance peaks of NTD-associated residues of [15]N-labelled R52E protein. In the presence of either $(GT)_6$ or $(GT)_{12}$ oligonucleotide, the chemical shifts for RRM1-2 residues involved in the binding with nucleic acids are similar for all the protein forms tested (Supplementary Fig. 6 and 7). Similar observations were made with GU-rich RNA repeats (Supplementary Fig. 8, see arrows). We can

thus conclude that NTD/NTD interactions mostly affect the cooperative binding but not the interaction of RRM residues with nucleic acids.

## Cooperativity weakens the NTD/NTD interface

We then focused on whether the residues associated with NTD/NTD interactions are affected by the binding to GT or GU repeats. NTD/NTD interactions lead to significant peak broadening of NTD residues, which affects their detection in the NMR spectra. A similar broadening was observed when the [15]N-labelled TDP-43 R52E mutant interacts with an unlabeled TDP-43 E17R mutant at equimolar concentration (Supplementary Fig. 9). The presence or absence of $(GT)_6$ or $(GT)_{12}$ did not significantly change this trend, which indicates that NTD/NTD interactions are preserved (Supplementary Fig. 10). However, an in-depth analysis of the chemical shifts and the heights of the resonance peaks revealed significant differences (Fig. 2A, B). Several NTD resonance peaks of wild type [15]N-labelled TDP-43 as well as of the equimolar mixture of [15]N-labelled TDP-43 R52E and unlabeled TDP-43 E17R reappear in the presence of $(GT)_{12}$ more markedly compared to $(GT)_6$ (Fig. 2B, residues: I18, S29, V31, T32, V74; Supplementary Fig. 9). Likewise, several NTD peaks also appear when wild type TDP-43 (a.a., 1-277) interacts with $(GU)_{12}$ RNA compared to $(GU)_6$ RNA (Supplementary Fig. 8, residues: I18, S29, T32, G37). Importantly, we observed larger chemical shift perturbations for resonances of several NTD residues located at the NTD/NTD interface in the presence of $(GT)_{12}$ compared to in the presence of $(GT)_6$ (E17, R55 and G53, Supplementary Fig. 11, Fig. 2A, B). In contrast, the resonance peaks of R52E NTD residues, unable to initiate NTD/NTD interactions, are always observable with no differences among the chemical shifts under all conditions. The chemical shifts of resonance peaks of NTD residues located at the NTD/NTD interface may indicate reduced NTD/NTD interactions in the presence of $(GT)_{12}$ or $(GU)_{12}$ compared to $(GT)_6$ or $(GU)_6$.

To more clearly probe NTD/NTD interactions, we performed competition assays using a truncated and unlabeled E17R NTD fragment (a.a., 1-93, lacking the RRM1-2 domains). E17R NTD does not bind to RNA but interacts with the NTD of R52E mutant (a.a 1-277) (Supplementary Figs. 10 and 12). Upon increasing concentrations of an unlabeled E17R NTD, the peaks of [15]N-labelled NTD residues related to the NTD/NTD interface become increasingly apparent and undergo large chemical shifts[18]. This phenomenon is due to the dynamical exchange between NTDs that enables the reappearance of the peaks of NTD residues. The lower the affinity between NTDs, the more pronounced the reappearance of NTD peaks should be. Here, we used a different version of this assay[18] (Fig. 2C, D). We added unlabeled E17R NTD (a.a., 1-93) in excess to compete the NTD/NTD interface involving [15]N-labelled TDP-43 R52E that initially interacts with an unlabeled TDP-43 E17R in the presence of $(GT)_{12}$ or $(GT)_6$. Interestingly, we observed the reappearance of NTD shifted peaks related to [15]N-labelled R52E with $(GT)_{12}$ and more significantly than with $(GT)_6$ (Supplementary Fig. 12). This probably means that the E17R NTD can access to the TDP-43 NTD interface more easily in presence of $(GT)_{12}$ than with $(GT)_6$. Furthermore, we then used a long oligonucleotide, $(GT)_6A_{12}(GT)_6$, in which a linker of 12 adenine nucleotides was added to prevent the cooperative association between adjacent TDP-43 along $(GT)_6A_{12}(GT)_6$, in agreement with ITC analysis (Supplementary Fig. 4C). In contrast with $(GT)_{12}$, $(GT)_6A_{12}(GT)_6$ does not enable the reappearance of TDP-43 NTD resonance peaks upon the addition of unlabeled E17R NTD fragment (a.a., 1-93) (Fig. 2C, D). The discontinuous binding of TDP-43 to $(GT)_6A_{12}(GT)_6$ thus secures NTD/NTD interactions.

In summary, NTD/NTD interactions between adjacent TDP-43 proteins are limited when TDP-43 binds cooperatively to long $(GT)_{12}$ repeats, unlike non-cooperative association to short or longer nucleic acids.

## NTD/NTD interactions compact long nucleic acids

To investigate structural changes on a larger scale than accessible by NMR spectroscopy, we performed SAXS analysis of in vitro-purified

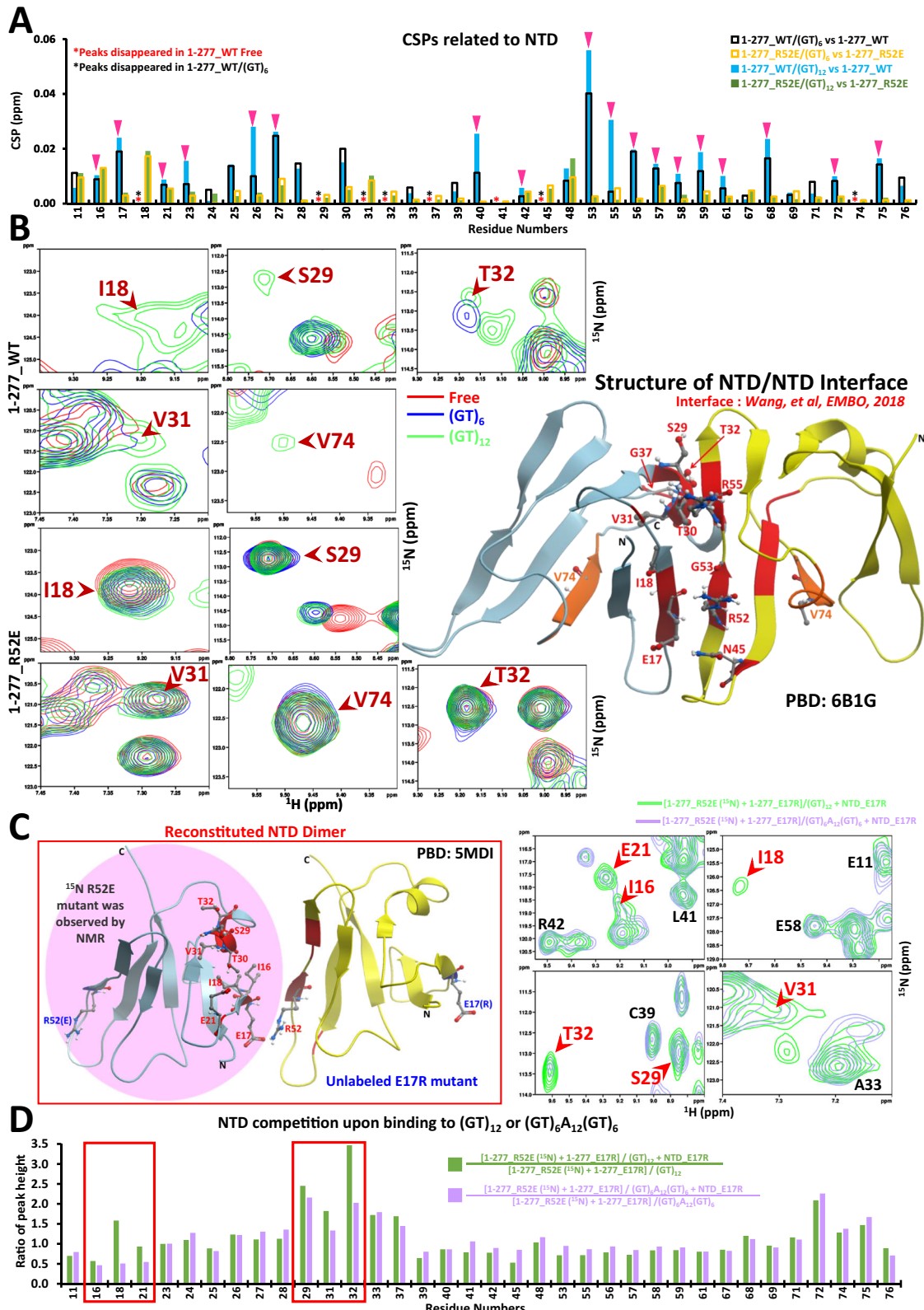

TDP-43:oligonucleotide complexes. We used four different protein combinations (a.a., 1-277) to explore the putative role of NTD/NTD interactions: the wild type TDP-43 and mutants R52E, E17R alone or together. The acquisition of the SAXS images was coupled to a Size Exclusion Chromatography (SEC) column (Supplementary Fig. 13A). The SEC chromatograms and the SAXS images were recorded during the elution. In the presence of $(GT)_{12}$ for which the binding is

cooperative (Fig. 1D), we observed that single mutants (blue and yellow trace) display similar elution volumes as WT (red trace) (Supplementary Fig. 13A). However, NTD/NTD interactions-deficient mutants display lower elution volumes due to a less compacted conformation than wild type TDP-43 in the presence of $(GT)_6A_{12}(GT)_6$, which cannot allow a continuous binding of TDP-43 because of the stretch of $A_{12}$ present in the middle of the oligonucleotide.

**Fig. 2 | The cooperative binding of TDP-43 to nucleic acids limits NTD/NTD interactions. A** CSPs of NTD residues that are not overlapped in 2D spectra under indicated conditions. Pink arrows indicate NTD/NTD interface residues for which significant CSPs are observable upon wild type TDP-43 binding to $(GT)_{12}$. **B** Left panel: NTD residues resonances that appear upon interactions with $(GT)_{12}$ compared to $(GT)_6$, but for wild type TDP-43 only. Right panel: Structure of an NTD dimer from Wang et al. (2018)[18]. The structure is obtained from the Protein Data Bank (PDB) database with code: 6B1G and can be used for free. The residues reported to participate into intermolecular NTD/NTD interactions are highlighted in red. **C**. Schematic view on NTD/NTD interactions based on structural data[62]. The structure is obtained from the PDB database with code: 5MDI and can be used for free. E17R and R52E cannot interact with themselves but only with each other to form a dimer R52E:E17R. Only R52E mutant is labelled with $^{15}N$, so changes of only one side of NTD/NTD interface can be detected. **D** Lower panel: Histograms show the ratio of NTD peak heights of R25E (a.a., 1-277) after versus before competition with unlabeled E17R NTD 24 times in excess (see material and methods). NTD residues involved in NTD/NTD interactions (Red boxes) are more sensitive to the competition with $(GT)_{12}$ compared with $(GT)_6A_{12}(GT)_6$. Upper right panel: Zoom in on some residues located in the NTD/NTD interface.

The SEC-SAXS data allowed us to estimate the radius of gyration ($R_g$) and the maximum distance ($D_{max}$) of the nucleoprotein complexes (Fig. 3, Supplementary Figs. 14 and 18). The results confirmed the observations based on the elution volumes. NTD/NTD interactions do not induce the compaction of nucleoprotein when TDP-43 is cooperatively associated to $(GT)_{12}$. Indeed, when $(GT)_{12}$ was used, the sizes of the nucleoprotein complexes are nearly identical for all the TDP-43 constructs, for wild type TDP-43 (162 Å, maximum diameter) alongside the range observed for the mutants (159 Å–165 Å), and therefore independent on the NTD/NTD interactions (Fig. 3A, Supplementary Fig. 13B, C and 15A). In addition, full atomic models of two TDP-43 bound to $(GU)_{12}$ were generated by using the DADIMODO web server (https://dadimodo.synchrotron-soleil.fr)[27]. These models were obtained by using a constructed model based on available 3D structure of NTD domain and a model of TDP-43 dimer from our previous investigations[25] and SAXS curves. In all the models, the two NTD domains are positioned far apart, which should limit NTD/NTD interactions. Here, the obtained models for the wild type as well as R52E mutant (a.a., 1-277) fit well the experimental SAXS curves with low discrepancy values ($\chi^2$) around 1.35–1.39 and 1.61–1.64, respectively (Supplementary Fig. 16).

To investigate the putative existence of a closed form displaying NTD/NTD interface between adjacent TDP-43 along $(GU)_{12}$, we applied distance constraints (4–5 Å) between R52 residue from NTD of one TDP-43 unit, and E17 residue from NTD of the second TDP-43 unit. Interestingly, the best-fitting models display significantly elevated $\chi^2$ values, 3.12 or 3.60 when R52 (from the first unit) interacts with E17 (from the second unit) or vice versa (E17/R52), respectively (Supplementary Fig. 17). Then, the best-constrained conformation was subsequently used as the starting point for a new calculation performed without any distance constraints by using DADIMODO program. Strikingly, the result of this calculation indicates that the best conformation obtained, with a $\chi^2$ value of 1.33 similar to the initial calculations, corresponds to an open model in which the two NTD domains are well separated by approximately 72 Å (Supplementary Fig. 17). In conclusion, all of the best generated solutions displayed the models of TDP-43 dimer cooperatively associated to $(GU)_{12}$ where the two adjacent NTDs are separated. Moreover, the closed conformation (harboring the NTD/NTD interface) gives a worse fit than the open state.

In contrast, in the case of $(GT)_6A_{12}(GT)_6$ oligonucleotide, with which the cooperative association of TDP-43 is not possible, we observed smaller $D_{max}$ values, indicating compaction of the nucleoprotein complexes when NTD/NTD interactions occur. This behavior is observed only for wild type TDP-43 ($D_{max}$: 166 Å) and for an equimolar mixture of E17R and R52E ($D_{max}$: 168 Å) (Fig. 3B and Supplementary Fig. 13B, C and 15 A). In addition, the complexes formed in the presence of $(GT)_6A_{12}(GT)_6$ remain extended with R52E ($D_{max}$: 203 Å) or E17R ($D_{max}$: 200 Å), when they are used alone. Similar results were obtained in the presence of GU RNA repeats oligonucleotides (Supplementary Fig. 14A, B and 15B). Therefore, as long as TDP-43 proteins are not positioned adjacently along the long GT- or GU-rich nucleotides through a cooperative association, TDP-43 NTD/NTD interactions could promote intermolecular bonds to compact nucleoprotein complexes. To further probe this model, the number of GT repeats was

progressively increased from 9 to 24 (Fig. 3C, Supplementary Figs. 18 and 19). We then measured the evolution of the $D_{max}$ in the presence of WT or R52E. We noticed a steady increase in $D_{max}$ versus the number of GT repeats, in absence of NTD/NTD interactions (R52E mutant). In contrast, when NTD/NTD interactions are possible, we observed no further increase in $D_{max}$ for either wild type TDP-43 or a mix of R52E and E17R, in the presence of the longest oligonucleotides $(GT)_{24}$. These results indicate that very long GT repeats complexes can bend on themselves because of long-ranged NTD/NTD interactions between non-adjacent TDP-43 (Fig. 3C).

Because of limited NTD/NTD interactions among adjacent TDP-43 proteins, the availability of NTD for interactions between NTDs located on distant clusters of GT repeats can be strongly promoted. NTD/NTD interactions and cooperativity would then be compatible to mutually reinforce instead of opposing each other. To document this point, we performed protein cross-linking assays in the presence of $(GT)_{12}$ as control and $(GT)_{12}A_6(GT)_{12}A_6(GT)_{12}$, the latter corresponds to 3 distinct GT clusters separated by 6-nt long adenine linkers (Fig. 3D). Consistently with the NTD-mediated compaction, cross-linking assay reveals the formation of tetramers of wild type TDP-43 complexed with $(GT)_{12}A_6(GT)_{12}A_6(GT)_{12}$ that are not present with R52E or with $(GT)_{12}$ ssDNA. The presence of tetramers most likely results from long-ranged NTD/NTD interactions that promote the formation of molecular bridges between two GT-rich clusters, each containing two units of TDP-43.

## NTD/NTD interactions and cooperativity interplay in cells

To assess the nature of TDP-43-associated intermolecular interactions in a cellular context, we used MT bench assays. In this assay, microtubules are used as intracellular platforms to detect RNA:RBP interactions in a cellular context but also the mixing:demixing between RBPs in mRNA-rich compartments that are formed following the expression of two different RBPs. We have previously taken advantage of this approach to reveal TDP-43 residues that are critical for its cooperative association with mRNA but also to analyze TDP-43 homotypic interactions[28]. Briefly, two different RBPs labeled with two different fluorophores (RFP and GFP) are brought onto microtubules through a fusion with a microtubule-binding domain (MBD). In each compartment, the enrichments of RFP- and GFP-labeled proteins are then measured. The measurement of $R^2$, the degree of linear correlation between the enrichment of the two RBPs in the compartments provides a mixing score between two RBPs (Fig. 4A, B). The mixing is low ($R^2 \ll 1$) when proteins form distinct RFP- or GFP-labeled compartments. The automatic acquisition of images by a high-resolution HCS imager allows obtaining robust mixing scores in 96-well plates over a large number of cells via an entirely automated pipeline (Fig. 4A, Supplementary Fig. 21A and Supplementary Fig. 21B for experimental replicates).

To probe the self-assembly of TDP-43 using MT bench assays, we prepared a combination of single- or double-point mutations. As previously demonstrated, G146A prevents the cooperative binding of TDP-43 to mRNA, but G146A does not alter TDP-43 structure nor TDP-43 affinity for short RNAs, according to NMR and ITC analyses[25]. R52E and E17R are two mutations preventing NTD/NTD interactions that we

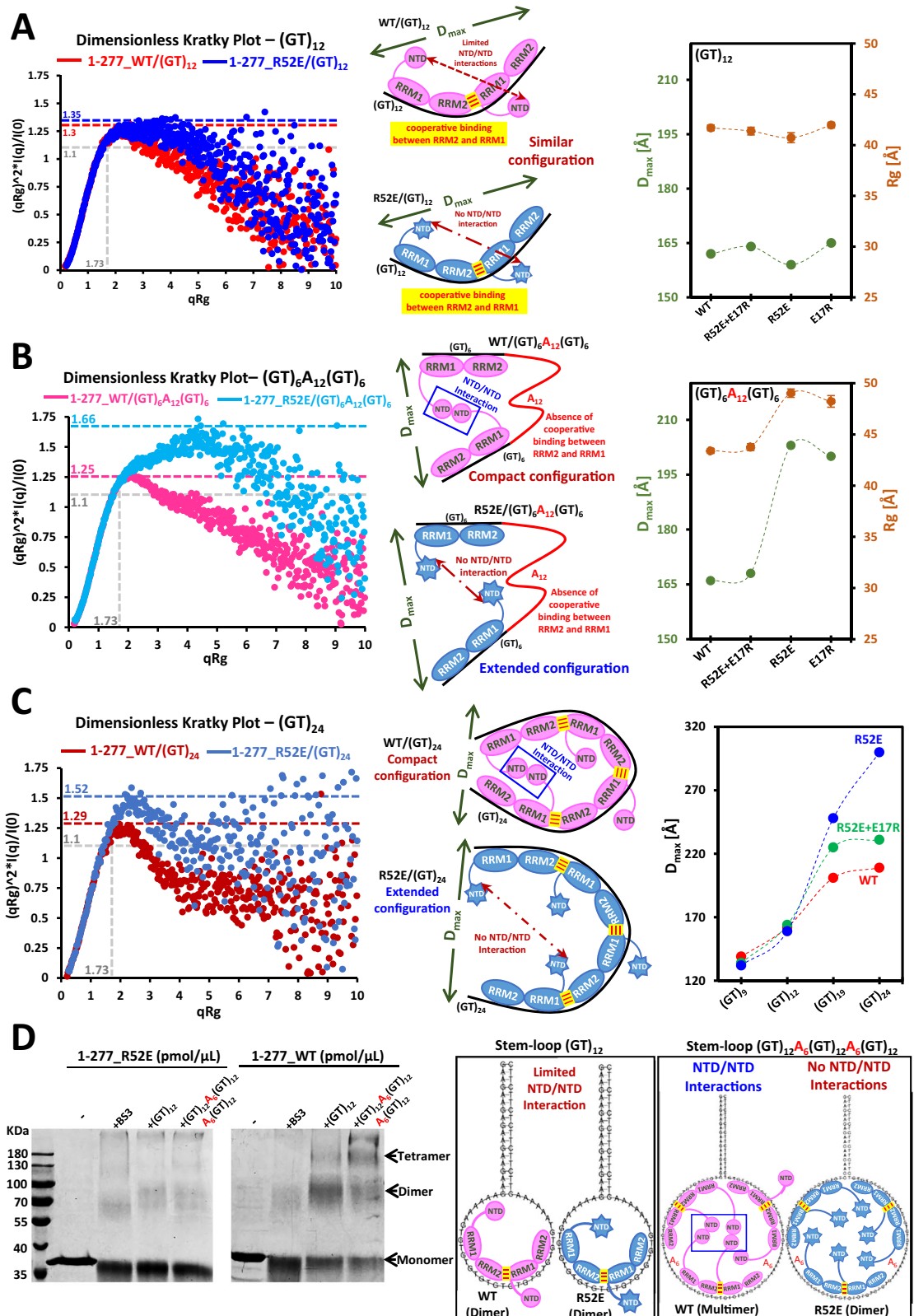

already used and validated (Fig. 1D). However, the co-expression of these two mutants allows the reconstitution of NTD/NTD interactions leading to form a dimer involving the non-mutated face of each complementary mutants E17R and R52E. The mutated face is ineligible to form NTD/NTD interactions.

R52E mutation is sufficient to impair the mixing with wild type TDP-43 (Fig. 4C) because each wild type unit possesses two complementary NTD/NTD interfaces and can form a multimer. However, E17R or R52E NTD mutants respectively preserve only one NTD/NTD interface, which is complementary to one of the wild type NTD faces. As expected, R52E mixes well with E17R to form a dimer. R52E displays a significant demixing with wild type (-15.3%) since R52E interacts with wild type NTD through only one binding site in contrast with wild type NTD.

**Fig. 3 | SAXS reveals the compaction of oligonucleotides due to NTD/NTD interactions. A** Left panel: Dimensionless Kratky Plots of wild type TDP-43 or R52E mutant in the presence of $(GT)_{12}$ oligonucleotides. Dotted lines in gray show the maximum for an ideal, compact, globular complex[63]. The complex formed with wild type or R52E is more extended than expected for a purely globular complex. Middle panel: Schematic view on TDP-43:$(GT)_{12}$ complexes. Right panel: The plot shows the radius of gyration, $R_g$, and the maximum dimension, $D_{max}$, of indicated TDP-43:$(GT)_{12}$ complexes. Data are presented in Supplementary Table 2. **B** Left panel: Dimensionless Kratky Plots of wild type or R52E TDP-43 in the presence of $(GT)_6$-$A_{12}$-$(GT)_6$ oligonucleotides. $(GT)_6$-$A_{12}$-$(GT)_6$ complex formed with wild type TDP-43 is less extended than with R52E. Middle panel: Schematic view on TDP-43:$(GT)_6A_{12}(GT)_6$ complexes. Right panel: The plot shows the radius of gyration, $R_g$, or the maximum dimension, $D_{max}$, of indicated TDP-43:$(GT)_6A_{12}(GT)_6$ complexes. Data are presented in Supplementary Table 2. **C** Left panel: Dimensionless Kratky Plots of wild type or R52E TDP-43 in presence of long $(GT)_{24}$ oligonucleotides. $(GT)_{24}$ complex formed with wild type TDP-43 is less extended than with R52E. Middle panel: Schematic view on TDP-43:$(GT)_{24}$ complexes. Right panel: Variation of $D_{max}$ values with increasing number of GT repeats. Wild type and the equimolar mix R52E + E17R but not R52E compact long oligonucleotides [$(GT)_{19}$ and $(GT)_{24}$]. **D** Left panel: SDS-polyacrylamide gel electrophoresis of samples from cross-linking experiments. Wild type TP-43 but not R52E generates the formation of tetramer and to much higher extent in the presence of $(GT)_{12}A_6(GT)_{12}$ than $(GT)_{12}$ (see arrows). Another experiment was performed with a longer loop (Supplementary Fig. 20). Right panel: Schematic view of the oligonucleotides used in the cross-linking assays and the complexes formed under the indicate conditions. Blue box: Example of a tetramer formation (two NTD dimers and two cooperative associations).

Furthermore, as expected, the double mutant, E17R-R52E, markedly reduces the mixing with wild type TDP-43 because the double mutation impairs NTD/NTD interactions. Interestingly, the double mutant G146A-R52E further decreases the mixing with wild type TDP-43 compared to G146A or R52E alone (-26.3% versus -11.2% and -13.0 %, respectively, Fig. 4D). Therefore, cooperativity and NTD/NTD interactions contribute incrementally to the mixing with wild type TDP-43 in compartments. Accordingly, the mixing of RFP-labeled G146A-R52E increases when GFP-labeled TDP-43 harbors either R52E or G146A mutation to reduce its capacity for self-assembly through NTD/NTD interactions or cooperativity, respectively (Fig. 4D, E).

Next, we explored the interplay between the cooperative binding of TDP-43 to mRNA and NTD/NTD interactions. To this end, we compared changes in mixing scores among cooperativity-deficient RFP or GFP-labelled G146A mutants (Fig. 4E, lower panel). We found that the double mutant G146A-R52E induces a strong demixing with G146A (-12.4%) comparable to cooperative-proficient mutant. These results indicate that NTD/NTD interactions occur in the absence of cooperativity. In contrast, among NTD-deficient RFP or GFP-labelled R52E, G146A mutation induces a significant but moderate demixing (-4%). Therefore, cooperativity occurs in the absence of NTD/NTD interactions, though but to a lesser extent than for wild type TDP-43 (Fig. 4E, upper panel).

Based on all these results, we propose a scheme for TDP-43 higher order assemblies consistent with the results obtained from MT bench assay (interplay between cooperativity and NTD/NTD interactions) and SAXS approach (NTD/NTD interactions are repressed for adjacent TDP-43) (Fig. 4F). We consider a cluster as multiple TDP-43 proteins bound along the same GU-repeats. From SAXS data, we know that NTD is available for inter- but not intra-cluster interactions. Therefore, NTD/NTD interactions should promote the recruitment of several TDP-43 from two distinct clusters to initiate multiple NTD/NTD interactions (Fig. 4F, upper panel). In turn, cooperativity may secure the formation of clusters (Fig. 4F, lower panel).

In summary, the analyses of the data obtained with the MT bench assay unravel that NTD/NTD interactions and TDP-43 cooperative association with mRNA each contribute to promote the formation of NTD-linked TDP-43 clusters.

## Cooperativity limits NTD-mediated RRM aggregation in cells

In one of the TDP-43 aggregation models, TDP-43-rich inclusions in cells appear following inter RRM interactions[26]. More precisely, under stress conditions, such as exposure to arsenite, acetylation of lysine residues leads to aggregation between RRMs[26]. Accordingly, in HeLa cells expressing HA-tagged TDP-43, we observed that even low concentrations of arsenite (80 µM for 1 h, Fig. 5A) are able to generate the formation of mRNA-free of detergent-insoluble TDP-43 aggregates (see reference[25]). Importantly, under the same conditions, when other RBPs harboring RRM domains such as HuR and FUS are expressed in cells in a similar manner to TDP-43, we did not observe any significant nuclear RBP aggregation as in the case of TDP-43 (Fig. 5B). These

results demonstrate the specific sensitivity of TDP-43 to oxidative stress. Next, using TDP-43 mutants that we have already characterized in vitro and with MT bench assays, we measured the formation of nuclear aggregates in cells expressing different mutants at similar levels to make a fair comparison (Supplementary Fig. 22B). When TDP-43 does not bind cooperatively to RNA (G146A mutant) very low arsenite concentration promotes a massive aggregation (down to 5 µM for 1 h, Fig. 5A) without sequestering mRNA, as expected for an RNA-free TDP-43 aggregation. The aggregation level is much more marked than with wild type TDP-43 (Fig. 5A). Interestingly, without NTD/NTD interactions, R52E mutation totally suppresses TDP-43 aggregation induced by arsenite (Fig. 5C). The double mutation, R52E-G146A, also completely suppresses the massive aggregation of G146A (Fig. 5C). Therefore, cooperativity antagonizes TDP-43 aggregation mediated by arsenite. In contrast, interactions between NTDs strongly promote the formation of TDP-43 aggregates. To further support these results, we analyzed the extraction of nuclear TDP-43 from cells treated with arsenite after short exposure to detergent (Supplementary Fig. 23). R52E-G146A and R52E mutants are better extracted than wild type TDP-43 and G146A, as expected if NTD/NTD interactions reduce TDP-43 solubility after arsenite treatment. In addition, we tested whether the nuclear TDP-43 aggregates are truly irreversible assemblies by washing out arsenite for 90 min (Supplementary Fig. 24). As control, stress granules which are known RBP-rich reversible condensates[29] mostly disappear after arsenite removal. In contrast, the presence of wild type and G146 A TDP-43 puncta in the nucleus persists, consistent with TDP-43 aggregation (or the formation of solid-like condensates with very low dissociation kinetics).

At the light of these results, we therefore performed a series of experiments to test the role of lysine acetylation within RRM1-2 in the aggregation of TDP-43. Two pairs of lysine residues, known to undergo acetylation following arsenite stress, were mutated to glutamine and arginine[26]. The glutamine substitution mimics the acetylation while the arginine substitution abolishes the ability to receive the acetylation modification. The double mutation, K136R-K137R, does not induce any noticeable consequence regarding the aggregation of TDP-43. On the other hand, the double mutation, K145R-K192R, prevents the aggregation of TDP-43 after arsenite exposure, even with the mutation G146A (Fig. 6A, B), while the double mutant, K145Q-K192Q, promotes the aggregation of TDP-43 (Supplementary Fig. 25A). Therefore, the absence of acetylation of these two residues simultaneously seems important because mutation of only one of these lysine residues is not sufficient to prevent the formation of aggregates (Supplementary Fig. 25B).

In view of the results of the arsenite-mediated aggregation assays, we propose the following mechanism of TDP-43 aggregation. When TDP-43 binds to mRNA in a non-cooperative manner, interactions between TDP-43 NTDs along RNA are then possible, as observed with cross-linking assays (Fig. 3D). Consequently, under stress conditions, there are still NTD/NTD interactions between nearby TDP-43 even after their release from mRNA. This NTD/NTD link between RNA-free TDP-43 increases the occurrence of interactions between acetylated RRMs

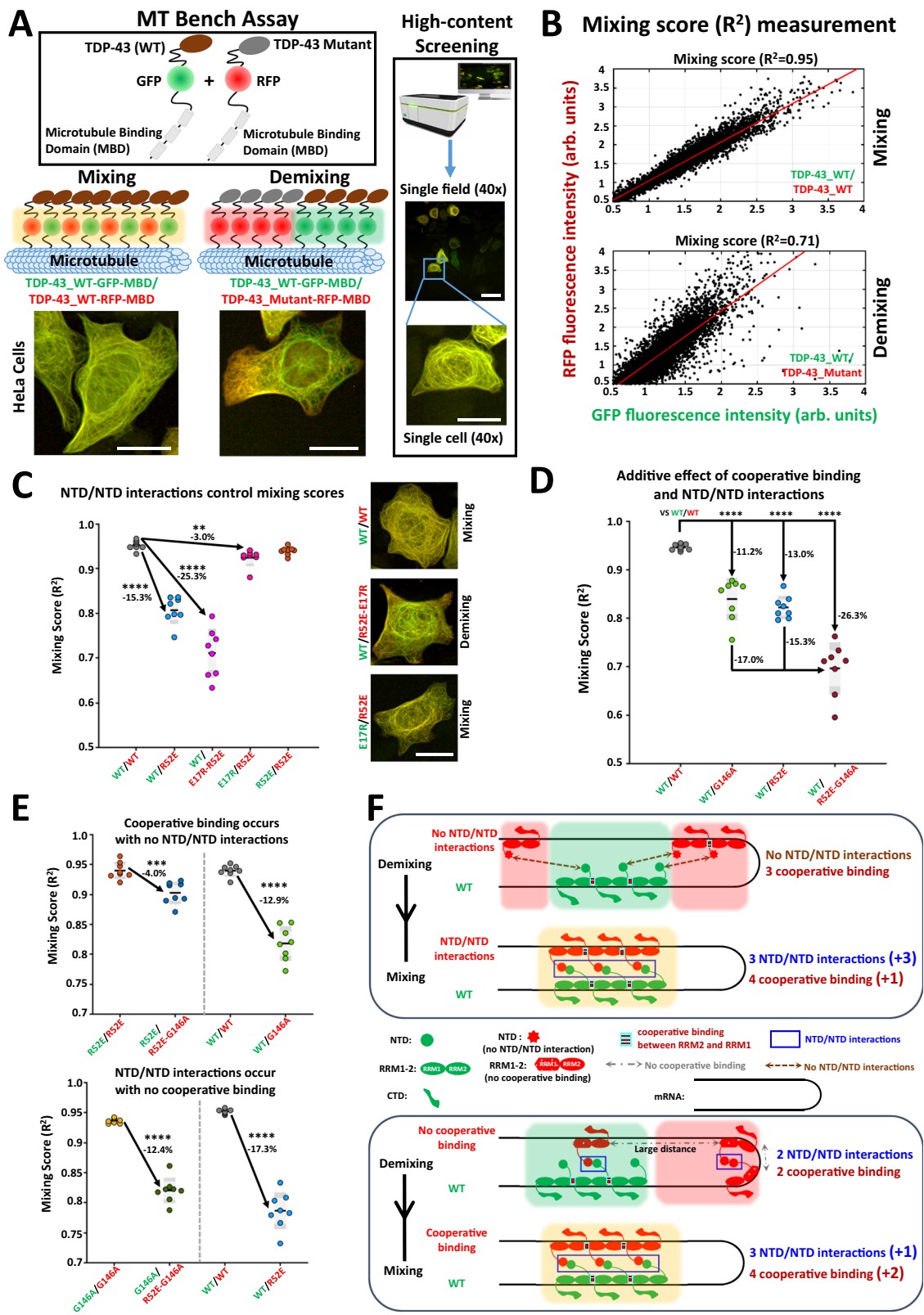

to promote aggregation. Long-ranged NTD/NTD links between distinct TDP-43 clusters under physiological conditions are less likely to increase the occurrence of RRM intermolecular interactions.

To further challenge this hypothesis, we used a well-characterized splicing reporter in which the presence of an intronic 13 GU repeats is recognized by TDP-43 to control the excision of exon 9 in the CFTR gene[30–32]. This construct has been previously validated since the

expression of TDP-43 is necessary to control the splicing of CFTR exon 9. We then modified this splicing reporter by adding a 7 nt-long adenine nucleotide linker right in the middle of the poly (GU) sequence. In this case, the cooperative binding of TDP-43 is no longer possible. Consistently, the splicing of CFTR exon 9, controlled by TDP-43, is impaired (Fig. 6C, Supplementary Fig. 26). After these different controls, we measured the aggregation of TDP-43 as a function of the

**Fig. 4 | Assessment of NTD/NTD interactions and the interplay with cooperativity in mRNA-rich compartments by using the microtubule bench assay.**
**A** Upper panel: Schematic view on the MT bench assays to probe the mixing/demixing between different TDP-43 constructs. The measurements were performed in 96-well plates in an entirely automated manner. Lower panel: Representative images of HeLa cells corresponding to mixing/demixing. Scale bar: 20 μm.
**B** Methods used to measure the mixing/demixing between two proteins. $R^2$, which is the degree of linear correlation, is used as a mixing score ($R^2$ -1, strong mixing and $R^2$ -0, strong demixing).
**C** Left panel: Mixing scores for indicated RFP/GFP-labeled TDP-43 couples co-expressed in HeLa cells. R52E and E17R have limited capacity to initiate NTD/NTD interactions. R52E-E17R can initiate NTD/NTD interactions with itself but not with wild type TDP-43. The asterisks indicate statistical significance

with $p < 0.01$, $p < 0.0001$, as measured by two-sided two-sample $t$-tests from $n = 8$ wells (each dot represents one well). Right panel: Representative images of HeLa cells. Scale bar: 20 μm. **D** Mixing score showing that the double mutation, R52E-G146A, further increments the demixing with wild type TDP-43 compared to single mutations (R52E or G146A). The asterisks indicate statistical significance with $p < 0.0001$, as measured by two-sided two-sample $t$-tests from $n = 8$ wells (each dot represents one well). **E** Interplay between cooperativity and NTD/NTD interactions as probed with indicated mutants. The asterisks indicate statistical significance with $p < 0.001$, $p < 0.0001$, as measured by two-sided two-sample $t$-tests from $n = 8$ wells (each dot represents one well). **F** Schematic view of the interplay between cooperative interactions and NTD/NTD interactions resulting in a different mixing score.

expression of splicing reporter pre-mRNA (GFP expression level). The aggregation of TDP-43 in the nucleus following exposure to arsenite continuously decreases with the expression level of the pre-mRNA containing 13 GU repeats (Fig. 6C). Conversely, increasing the expression of pre-mRNA containing $(GU)_6A_7(GU)_7$ does not limit nuclear TDP-43 aggregation. In summary, the results indicate that the cooperative association of TDP-43 prevents NTD/NTD interactions between adjacent TDP-43 to drastically decrease nuclear TDP-43 aggregation under stress conditions.

## Discussion

Under physiological conditions, TDP-43 binds to intronic sequences in the nucleus[25]. Unlike FUS that binds to intron nonspecifically resulting in a sawtooth-like pattern obtained by CLIP[33], TDP-43 transcriptomic data show a discrete pattern with a strong preference for GU-rich sequences[34]. These results were also confirmed in vitro since the RRM tandem of TDP-43, RRM1-2, has a high affinity for short GU sequences[35]. In addition, TDP-43 binds cooperatively along GU-rich RNA sequences through direct intermolecular interactions between RRM1 and RRM2 which secures a continuous binding of multiple TDP-43 along long GU-rich sequences[25]. Such long GU-rich sequences (>12 repeats and up to 30 repeats) are especially found in long introns with a high occurrence in neuronal cells[36–38]. Accordingly, it has been proposed that TDP-43 allows the processing of neuronal pre-mRNAs containing long introns to avoid unproductive splicing that would lead to mRNA degradation[36].

Our structural data indeed confirm that TDP-43 binds cooperatively to GU repeats, but, importantly, we show here that the spatial arrangement of TDP-43 along GU-rich mRNA prevents intermolecular interactions between NTDs from consecutive TDP-43 units (Fig. 3A, B). This is due to the cooperative association of multiple TDP-43 along the same GU-rich sequences, which keeps NTDs well separated from each other. NTDs are therefore free to interact with other TDP-43 units located in other GU-rich clusters (Fig. 7). The long-ranged TDP-43 bridges secured by NTD/NTD interactions between GU repeats clusters may allow the compaction of mRNA introns (Figs. 3C, D and 7b). This process may be very efficient given the availability of multiple NTDs aligned along each cluster to initiate NTD/NTD interactions. Consistently with this hypothesis, we observe the compaction of TDP-43 nucleoprotein complex by SAXS when we inserted several adenosine nucleotides in the middle of GT repeats to break the cooperative binding of TDP-43 (Fig. 3B). The importance of NTD/NTD interactions in the compaction of introns may also explain the critical role of NTD in TDP-43-related splicing events reported in the literature[18,23]. The structural insights obtained in this study may therefore provide further insights into the specific role of TDP-43 in mRNA splicing by introducing TDP-43 as a distinct RBP to compact long mRNA introns. Furthermore, the role of the low complexity domain (LCD) at the C-terminus of TDP-43 was not investigated in this work due to solubility reasons. The C-terminal LCD of TDP-43 carries most of the pathological mutations and could provide another level of regulation

of TDP-43:mRNA nucleoprotein complexes, notably through post-translational modifications. Both aspects deserve further investigation in the context of our model.

Then, to get further insights into the mechanisms leading to the pathological aggregation of TDP-43, we took advantage of having a better view on the physiological TDP-43 assemblies with GU-rich sequences. RNA is already known to prevent aberrant phase separation of RBPs[26,39], including for TDP-43 when it is associated to GU-rich sequences[25,40]. Under physiological conditions, TDP-43 proteins are bound cooperatively along GU-rich intronic sequences, which prevents intermolecular NTD/NTD links between adjacent TDP-43. Therefore, when TDP-43 transiently dissociates from mRNA, as expected for dynamic interactions, TDP-43 is free to diffuse away in the absence of intermolecular NTD/NTD links. In contrast, non-cooperative associations of TDP-43 proteins to RNA promote NTD/NTD intermolecular links between adjacent proteins (Fig. 3B). In this case, when TDP-43 proteins are released from RNA, NTD/NTD links may increase the occurrence of intermolecular RRM interactions (Fig. 7). Importantly, TDP-43 RRMs is particularly prone to RRM aggregation since oxidative stress causes the acetylation of lysine residues which leads to irreversible RRM assemblies, as already documented[26,41]. Within the frame of this model, the cooperative binding of TDP-43 to GU-rich mRNA appears central to preserve the solubility of TDP-43 RRMs under oxidative stress conditions, which are conditions known to be experienced by ageing neurons in neurodegenerative diseases[42,43]. In agreement with this notion, we show that the nuclear aggregation of TDP-43 mediated by arsenite through the self-association of tandem RRMs is drastically increased for cooperativity-deficient mutants if and only if NTD/NTD interactions occur (Fig. 5C). Similarly, the nuclear expression of pre-mRNAs with GU-repeats in HeLa cells antagonizes the aggregation of wild type TDP-43, which is not the case when an insertion of an adenosine-rich linker prevents the cooperative association of TDP-43 to RNA (Fig. 6C).

Based on these results, we anticipate that NTD/NTD interactions could play a major role in the formation of cytoplasmic aggregates. Indeed, in the cytoplasm, GU-rich sequences are present in limited numbers because they are mainly found in introns of nuclear pre-mRNAs. This point could provide a rational explanation for the preponderance of cytoplasmic inclusions compared to nuclear aggregates in neurons of ALS patients. However, an opposite view may be also consistent with this model since NTD/NTD interactions are important to retain TDP-43 in the nucleus[44], likely by forming NTD/NTD bridges between GU repeats. In this case, weakened NTD/NTD interactions may promote the translocation of TDP-43 in the cytoplasm[20,44], which may in turn have a negative impact on neuronal functions[45]. It would be interesting to document the relative weights of these opposite contributions to TDP-43 aggregation in ageing neurons.

Altogether, the results provide an interesting working hypothesis about the physiological role of TDP-43 in the processing of long introns that deserves to be further explored. However, whether compacting long introns help for the processing of mRNA splicing in cells is

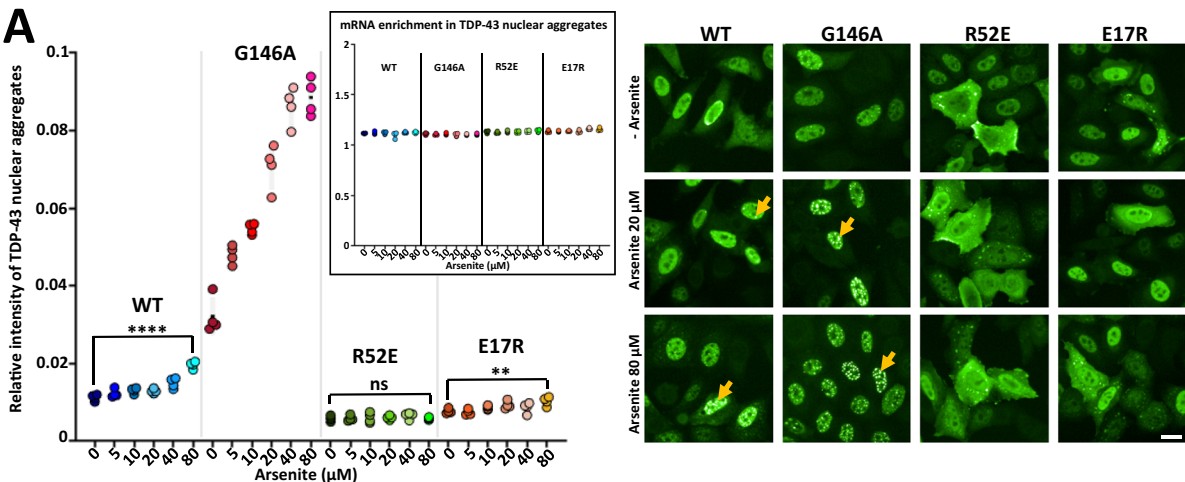

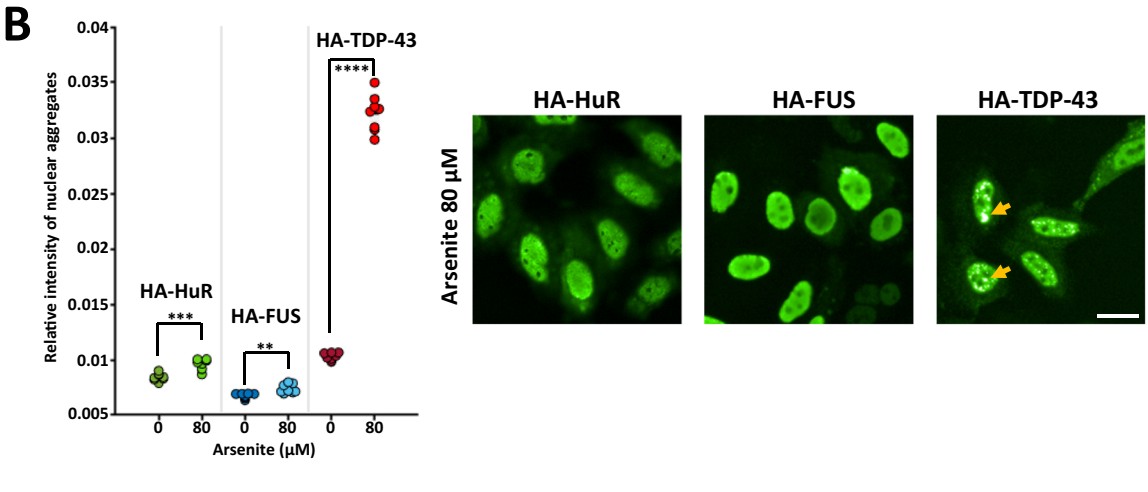

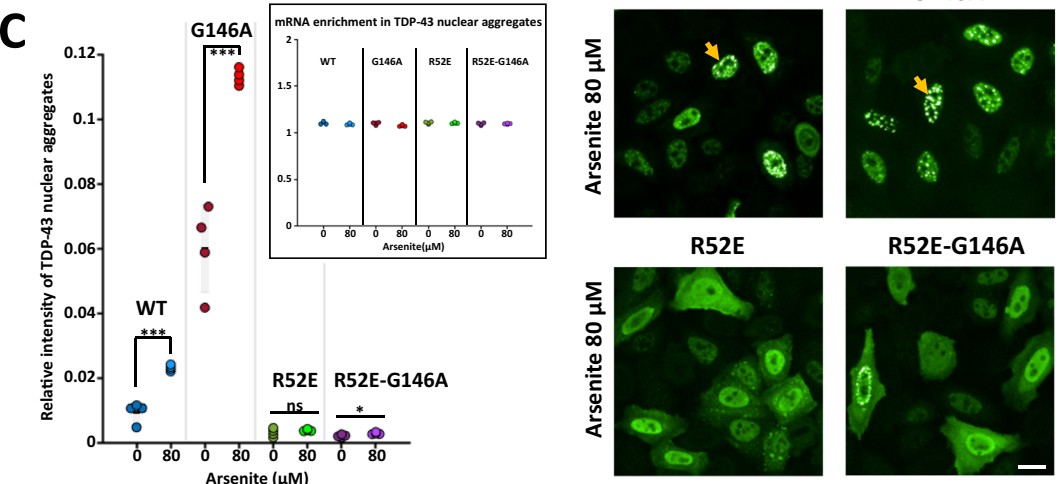

an open question. The data also show that the cooperative association of TDP-43 is critical to limit NTD-mediated TDP-43 aggregation. Both phosphorylation, ubiquitination, acetylation but also some pathological mutations such as P112H[46], D169G[47] and K263E[48] occur inside or in the vicinity of RRM1-2. To which extent these modifications affect, the cooperativity also deserves to be explored. In addition, unlike the purely homotypic interactions generated by the TDP-43 NTD, interactions between LCDs may involve heterotypic interactions with other RBP partners[49] and can be modified by oxidation[50]. Determining the role of LCD heterotopic interactions and post-transitional modifications in the higher-order assembly of TDP-43 in the presence or absence of GU repeats may therefore be of interest to explore whether other RNA-binding proteins that are also associated with ALS or FTLD may play a role in this process.

**Fig. 5 | Non-cooperative binding to RNA leads to massive arsenite-mediated TDP-43 aggregation. A** Left panel: The HA-relative intensity of nuclear aggregates versus surrounding nucleoplasm was measured to score TDP-43 aggregation in HeLa cells expressing HA-tagged different TDP-43 constructs under indicated arsenite concentrations. Automated measurement from cells grown in 96 well plated. One dot represents the mean value in one well. The asterisks indicate statistical significance with ns, non-significant, $**p < 0.01$, $****p < 0.0001$, as measured by two-sided two-sample $t$-tests from $n = 4$ wells. Inset: mRNA enrichment in TDP-43 aggregates. No increase in mRNA enrichment is observed with all constructs. Right panel: Representative images (HA-antibody). Scale bar: 20 μm. **B** Left panel: TDP-43 aggregation score under indicated conditions in cells expressing HA-tagged HuR, TDP-43 or FUS,

different RRM harboring proteins. The asterisks indicate statistical significance with $**p < 0.01$, $***p < 0.001$, $****p < 0.0001$, as measured by two-sided two-sample $t$-tests from $n = 8$ wells (each dot represents one well). Right panel: Representative images (anti-HA). Scale bar: 20 μm. **C** Left panels: TDP-43 aggregation score under indicated conditions in cells expressing HA-tagged TDP-43 constructs. Inset: mRNA enrichment in TDP-43 aggregates. No increase in mRNA enrichment is observed with all constructs. The asterisks indicate statistical significance with $*p < 0.05$, $***p < 0.001$, ns non-significant, as measured by two-sided two-sample $t$-tests from $n = 4$ wells (each dot represents one well). Right panel: Representative images (anti-HA). Scale bar: 20 μm.

## Methods

### Protein production and purification

The recombinant His$_6$-tagged NTD fragment, His$_6$-tagged RRM1-2 fragment and His$_6$-tagged NTD-RRM1-2 fragment of human TDP-43 were carried by pET plasmids (Supplementary Table 1).

Point mutations were introduced in pTDP-NTD_1-93 and pTDP-NTD-RRM1-2_1-277 expression plasmids by the 'Quikchange II XL site-directed mutagenesis kit' from Stratagene and corresponding oligonucleotides (Eurofins Genomics). DNA sequencing (Eurofins Genomics) checked all mutant sequences.

Proteins were overexpressed in *E. coli* strain BL21 (DE3). Cells grew at 37 °C in 2YT-kanamycin medium for unlabeled proteins or in minimal medium M9 supplemented with $^{15}NH_4Cl$ for $^{15}N$-labeled proteins. IPTG, which need to be a final concentration of 1 mM, was added into the culture when the optical density of the culture reached 0.8. IPTG is a reagent for efficiently initiating the transcription of lac operon to express the target gene in the operon. The bacteria grew continually for about 3.5 h.

Bacteria were gained and washed with 25 mM Tris-HCl buffer (washing buffer) at pH 7.6, containing 0.5 mM DTT and 2 M KCl. The pellet was suspended with about 20 mL-35 mL of 25 mM Tris-HCl buffer (lysis buffer) at pH 7.6, containing 0.5 mM DTT, 1 mM PMSF, EDTA-free protease inhibitor Cocktail (Roche) and 2 M KCl, and broken by sonication on ice (Bioblock Vibracell sonicator, model 72412). The final mixture was centrifuged for 30 min at $74,000 \times g$ 4 °C in a TL100 Beckman centrifuge. The supernatant was collected for protein purification.

The collected supernatant was purified following the manufacturer's recommendations (Qiagen). 7.5 mM imidazole was added into the collected supernatant and the supernatant was incubated overnight at room temperature with Ni$^{2+}$-NTA agarose (Qiagen) (20 mg of proteins/mL of resin) which has already pre-equilibrated in washing buffer. After incubation, the protein can bind to the resin via Histidine to form the polymer. The polymer was transferred to an Econo-Pac chromatography column (Bio-Rad) and then washed extensively with lysis buffer containing 10 mM imidazole. After that, the polymer was washed with washing buffer containing 15 mM imidazole. Then, the elution of the protein was obtained by increasing the concentration of imidazole from 20 mM to 250 mM, in 25 mM Tris-HCl buffer (buffer A) at pH 7.6, containing 0.5 mM DTT and 500 mM KCl. Thermo Scientific™ NanoDrop™ One/OneC Microvolume UV-Vis Spectrophotometer detected the concentration of the elution of protein. The fractions (usually from 50 mM to 250 mM imidazole) of pure protein were pooled and incubated with the His$_6$-tagged TEV protease to cut the His$_6$-tag peptide from the target protein. The protease was mixed with the target protein (0.5 μM TEV to ~30 μM protein) in buffer A, containing 1 mM DTT and 1 mM EDTA. The digestion was overnight at room temperature. A PD-10 column (GE Healthcare) was used for imidazole removing and buffer exchange. The TEV protease and His$_6$-tag peptides from the target protein were caught on the Ni$^{2+}$-NTA agarose column, and the target protein was collected in the pass-through fractions (the target protein are not bound to Ni$^{2+}$-NTA agarose).

All purification steps were monitored by using SDS-PAGE to verify that the desired TDP-43 fragments were finally obtained. To verify the NTD-RRM1-2 fragment of TDP-43, 12% resolving gel was used, while 15%

resolving gel was used for the RRM1-2 fragment and the NTD fragment verification.

Purified proteins were stored in 25 mM Tris-HCl buffer at pH 7.6, containing 1 M KCl and 0.5 mM DTT, at -80 °C.

### Isothermal titration calorimetry measurements

ITC experiments were performed at 25 °C by a MicroCal PEAQ-ITC isothermal titration calorimeter (Malvern Instruments). All the samples were prepared in 20 mM Hepes buffer at pH 7.6, containing 100 mM KCl and 2 mM TCEP. The protein concentration in the microcalorimeter cell (0.2 mL) varied from 13 μM to 18 μM. 37 injections of 1 μL (or 39 injections of 1 μL) of oligonucleotide solution [(GT)$_6$ at concentration 200 μM, (GT)$_{12}$ at concentration 100 μM and (GT)$_6$A$_{12}$(GT)$_6$ at concentration 100 μM] were carried out at 90 s intervals, with stirring at 750 rpm. Theoretical titration curves were fitted to the experimental data using the MicroCal PEAQ-ITC Analysis Software provided by the manufacturer. This software models the relationship between the heat released during each injection and several parameters: ΔH (enthalpy change, in kcal·mol$^{-1}$), K$_D$ (dissociation constant, in M), $n$ (number of binding sites per monomer), as well as the total protein concentration and the concentrations of free and total oligonucleotide. Changes in the free energy and entropy upon binding were calculated from equilibrium parameters obtained. Samples in the presence of (GT)$_6$ and (GT)$_6$A$_{12}$(GT)$_6$ were fitted with the one set of sites model, which enables the determination of the association constant $K$, the binding enthalpy ΔH, and the stoichiometry $n$ of the interaction. The cumulative heat $Q$ after each injection is given by Eq. (1):

$$Q = M_t V_0 [n \Delta H \frac{KX}{1 + KX}] \tag{1}$$

where $V_0$ is the cell volume, $M_t$ is the total protein concentration, and $X$ is the free oligonucleotide concentration in the active volume.

Samples in the presence of (GT)$_{12}$ were fitted with the two sets of sites model, enabling determination of the association constants ($K_1$, $K_2$), binding enthalpies (ΔH$_1$, ΔH$_2$), and site stoichiometries ($n_1$, $n_2$) for each binding site. The cumulative heat $Q$ after each injection is described by Eq. (2):

$$Q = M_t V_0 [n_1 \Delta H_1 \frac{K_1 X}{1 + K_1 X} + n_2 \Delta H_2 \frac{K_2 X}{1 + K_2 X}] \tag{2}$$

where $V_0$ is the cell volume, $M_t$ is the total protein concentration, and $X$ is the free oligonucleotide concentration in the active volume.

For both models, the heat observed for each injection, $q_i$, is calculated as shown below in Eq. (3):

$$q_i = \Delta Q_i = Q_i + \frac{dV_i}{V_0} [\frac{Q_i + Q_{i-1}}{2}] - Q_{i-1} \tag{3}$$

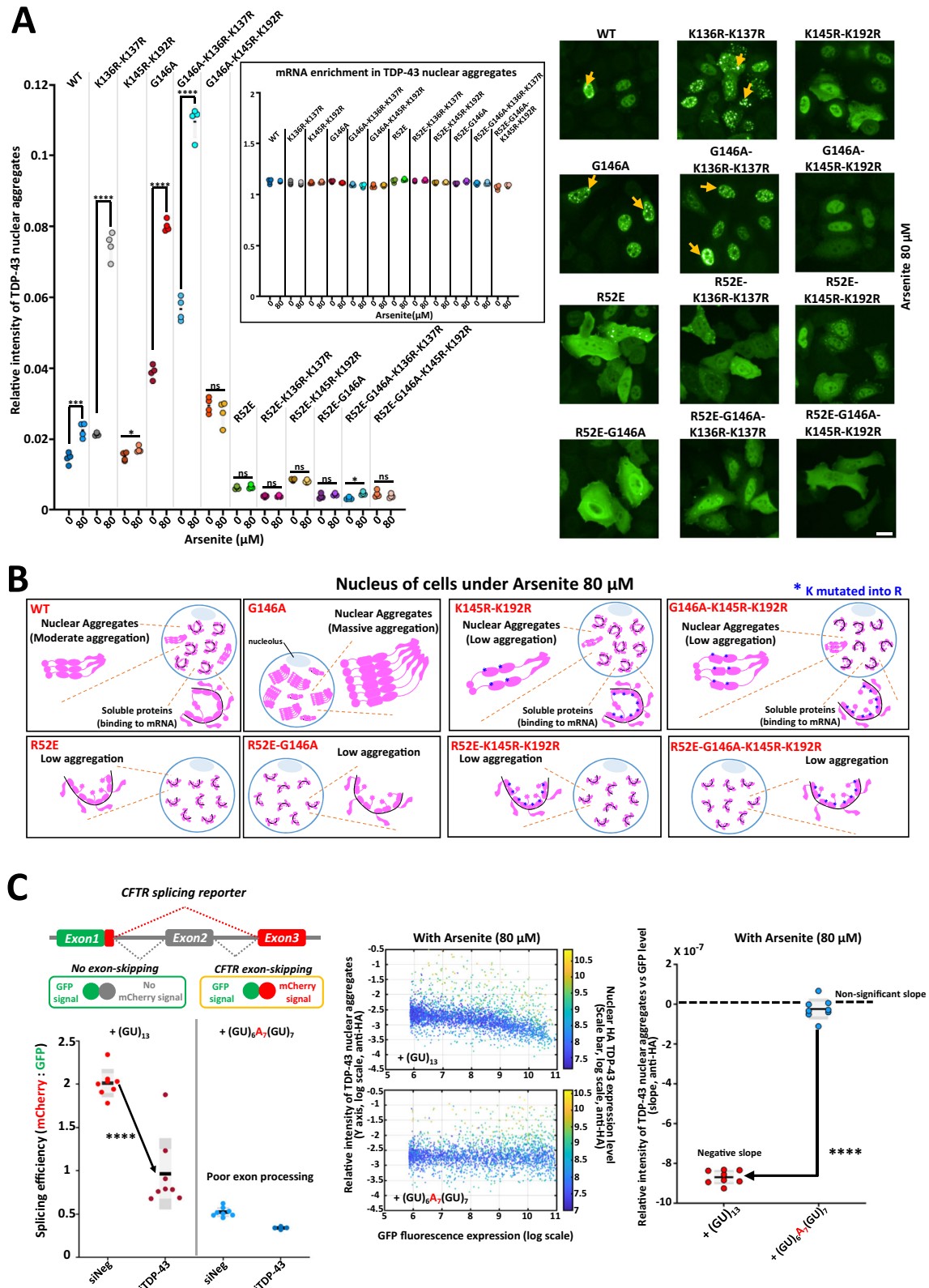

This approach follows respectively, the one set of sites and two sets of sites models formalism implemented in PEAQ-ITC and is based on established ITC methodology[51,52].

**Nuclear magnetic resonance of TDP-43 fragments**
For NMR samples preparation, 30 μM of purified proteins ($^{15}$N-labeled NTD-RRM1-2 fragments, NTD fragments or RRM1-2 fragments) were

prepared as free protein samples. For DNA oligonucleotide-bound protein samples, (GT)-rich DNA oligonucleotides [$(GT)_6$: 35 μM, $(GT)_6A_{12}(GT)_6$: 17.5 μM or $(GT)_{12}$: 17.5 μM] were heated at 95 °C for 1 min and incubated with 30 μM purified proteins. The same treatments were done for $^{15}$N-labeled NTD-RRM1-2 fragments with (GU)-rich RNA oligonucleotides [$(GU)_6$, $(GU)_6A_{12}(GU)_6$ or $(GU)_{12}$] to prepare RNA-bound protein samples.

**Fig. 6 | Acetylation of RRM1-2 residues promotes aggregation that is antagonized by a cooperative association to GU repeats. A** Left panel: Arsenite-mediated TDP-43 aggregation score under indicated conditions in cells expressing different HA-tagged TDP-43 mutants. Non acetylable K145R-K192R double mutation prevents TDP-43 aggregation. Inset: mRNA enrichment in TDP-43 granules. No increase in mRNA enrichment is observed. The asterisks indicate statistical significance with*$p < 0.05$, **$p < 0.01$, ***$p < 0.001$, ****$p < 0.0001$, ns non-significant, as measured by two-sided two-sample $t$-tests from $n = 4$ wells (each dot represents one well). Right panel: Representative images (anti-HA). Scale bar: 20 μm. **B** Schematic view of the role of RRM acetylation and TDP-43 cooperative association to RNA in TDP-43 aggregation triggered by arsenite stress. **C** Left upper panel:

Schematic view of the splicing reporter assays for TDP-43 dependent excision of exon 9 in the CFTR gene. Left lower panel: Arsenite mediated TDP-43 aggregation score in HeLa cells versus the expression level of the splicing reporter with $(GU)_{13}$ repeats or a broken $(GU)_6A_7(GU)_7$ repeat to prevent cooperativity. One dot is one well. The asterisks indicate statistical significance with ****$p < 0.0001$, as measured by two-sided two-sample $t$-tests ($n = 8$ wells in 96-well plate). Middle panel: Relative intensity of TDP-43 nuclear aggregates versus GFP fluorescence expression. Scale bar means nuclear TDP-43 expression level. Right panel: Measurement of the slope of the curve. The asterisks indicate statistical significance with ****$p < 0.0001$, as measured by two-sided two-sample t-tests. GU repeats expression limits TDP-43 aggregation.

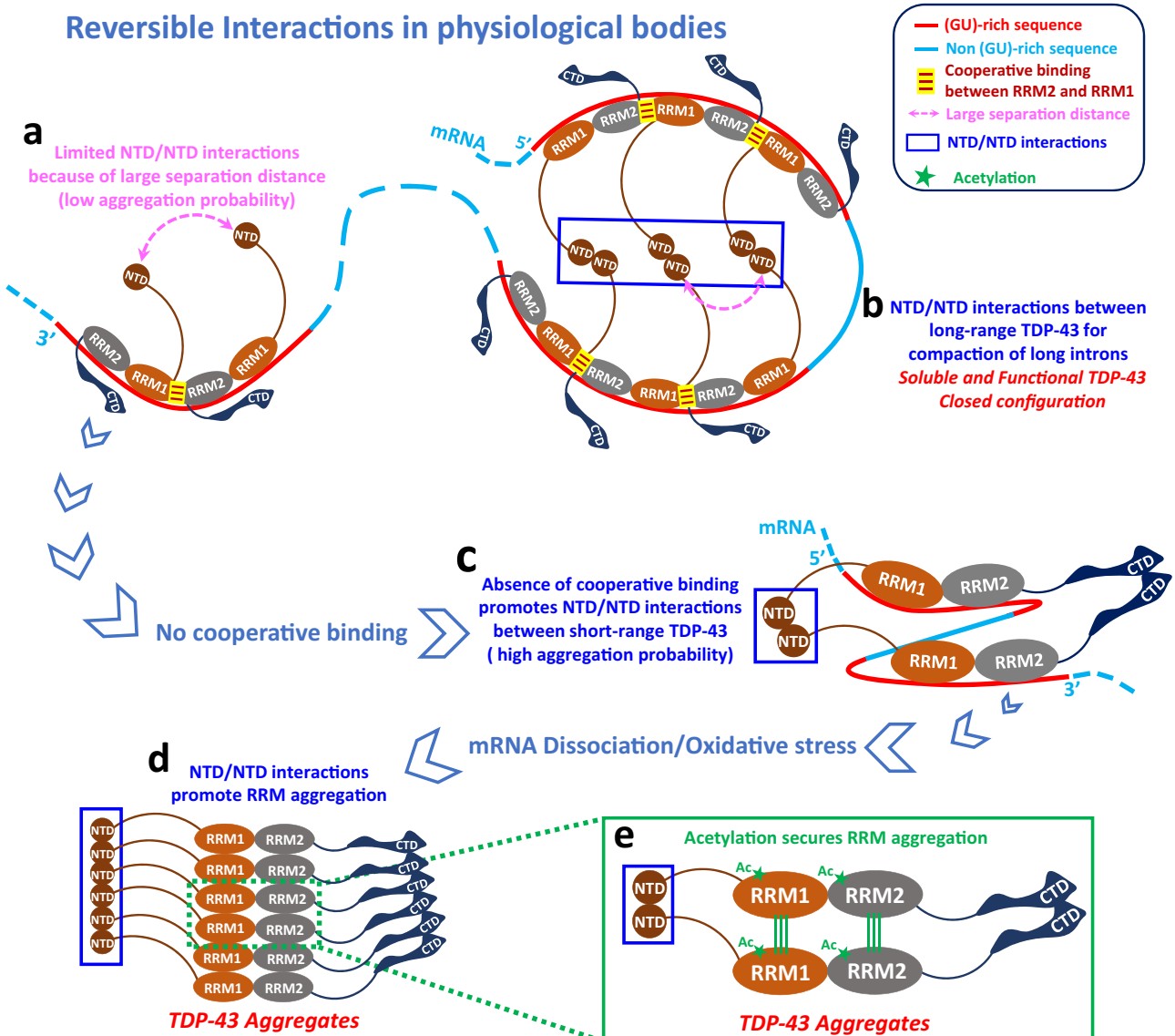

**Fig. 7 | Model representation of the higher order assemblies of TDP-43 in the presence of mRNA and the transition toward TDP-43 aggregation. a** Limited NTD/NTD interactions between cooperatively-associated TDP-43 along GU-rich RNA are due to the long separation distance between consecutive RRMs. **b** NTD/NTD interactions between distant clusters of TDP-43 proteins bound to GU-rich mRNA lead to an efficient compaction through arrays of NTD/NTD interactions,

owing to the cooperativity. **c** Uncooperative association to mRNA allows adherent NTD/NTD interactions between adjacently bound TDP-43 along mRNA that should promote inter-RRM interactions and therefore aggregation. **d** Classical view of inter-RRM and head-to-tail NTD assemblies giving the structure of RNA-free TDP-43 aggregates. **e** Zoom in the structure of RNA-free TDP-43 aggregates: RRM acetylation secures RRM aggregation.

In dimer reconstruction experiments, the mixture of proteins contained 30 μM 1-277_R52E (labeled) and 30 μM 1-277_E17R (unlabeled). The protein mixture was incubated with DNA oligonucleotides [$(GT)_6$: 70 μM, $(GT)_6A_{12}(GT)_6$: 35 μM or $(GT)_{12}$: 35 μM].

For NTD competition experiments, 30 μM 1-277_R52E (labeled) was mixed with 30 μM 1-277_E17R (unlabeled) and then incubated with DNA oligonucleotides [$(GT)_6$: 70 μM, $(GT)_6A_{12}(GT)_6$: 35 μM or $(GT)_{12}$:

35 μM]. The complex mixture was incubated with 720 μM (24 times to 1-277_E17R) 1-93_E17R (unlabeled isolated NTD) in final.

All the samples were prepared in 12 mM phosphate at pH 6.9, containing 100 mM KCl and 2 mM TCEP. RNA-bound protein samples were supplemented with SUPERase·In RNase Inhibitors (ThermoFisher Scientific). Samples were concentrated to a final volume of 180 μL, containing 10 μL D₂O for locking procedure and then transferred into 3 mm diameter NMR tubes (NORELL) for experiment acquisition. 2,2-Dimethyl-2-silapentane-5-sulfonic acid was used as a reference in pure D₂O (Eurisotop) for chemical shift referencing.

For NMR measurements, all the experiments were 2D SOFAST-HMQC experiments[53]. NMR spectra were acquired on a Bruker AVIII HD 600 MHz spectrometer equipped with a triple-resonance cryoprobe at 308 K and processed by Topspin 4.3.0 (Bruker). The number of scans and the relaxation delay corresponding to experiments involving in 1-277_WT or 1-93_WT were set to 1024 and 0.2 s. Experiments involving in mutants corresponded to a number of scans of 332 and a relaxation delay of 0.2 s. The number of scans and the relaxation delay of the experiments of 101-277_WT were 512 and 0.2 s. All the experimental data were acquired with 2048 and 128 complex pairs in direct ¹H dimensions and the indirect ¹⁵N. The corresponding acquisition time 136 ms (¹H) and 30 ms (¹⁵N) and the corresponding sweep widths were 12.5 ppm (¹H) and 34.5 ppm (¹⁵N). Shaped pulse length and power were calculated by considering an amide ¹H bandwidth of 4.5 ppm and a chemical shift offset of 8.5 ppm.

For NMR analysis, CcpNmr AnalysisAssign 3.1.0 was used for NMR spectra assignment and analysis. The chemical shifts obtained from our NMR experiments are similar to those available in the BMRB database, which facilitated the assignments. The ¹H and ¹⁵N chemical shift assignments of NTD-RRM1-2 fragments or RRM1-2 fragments were done by using published assignments for NTD and unbound RRM1-2 (BMRB Entries: 30345 and 27613, respectively). The ¹H and ¹⁵N chemical shift assignments of NTD-RRM1-2 fragments or RRM1-2 fragments binding to (GT)-rich or (GU)-rich oligonucleotides were performed by using published assignments for NTD (BMRB Entry: 30345) and (AUG12)-bound RRM1−2 (BMRB Entry: 19290). All these BMRB entries are published in the Biological Magnetic Resonance Bank (BMRB) database and can be used for free. The chemical shift perturbations (CSPs) were calculated by the formula following Eq. (4):

$$\Delta\delta = \left( 0.5 \left[ (\Delta\delta_H)^2 + (0.14\Delta\delta_N)^2 \right] \right)^{0.5} \tag{4}$$

where $\Delta\delta_H$ and $\Delta\delta_N$ are chemical shift variations in proton and nitrogen dimensions, respectively)

## Small-angle X-ray Scattering (SAXS)

SAXS data were collected on SWING beamline at Synchrotron SOLEIL (Saint- Aubin, France)[54]. Data collection parameters are shown in Supplementary Table 2. All the measurements were performed in the SAXS measuring cell with a 1.5 mm diameter quartz capillary. The measurements of NTD-RRM1-2 fragments in presence of RNA or DNA oligonucleotides were performed by using an Agilent BioSec5-500 column. The concentration of proteins and oligonucleotides were reported in Supplementary Table 2. All the samples were prepared in 20 mM Hepes buffer at pH 7.6, containing 100 mM KCl and 2 mM TCEP. The samples in presence of RNA were supplemented with SUPERase·In RNase Inhibitors. In each experiment, 50 μL of sample was loaded into the column with a flow rate fixed at 0.3 mL/min at 25 °C. Scattering of the elution buffer before void volume was recorded and used as buffer scattering for subtraction from protein/oligonucleotide patterns.

2D SAXS images obtained were radial averaged and normalized by Foxtrot, a Swing in-house software, and the 1D curves were analyzed by

BioXTAS RAW 2.2.1[55]. An advanced Series processing in RAW−Regularized Alternating Least Squares (REGALS)[56] was also used for some samples to decompose when peaks overlapping occurred to eliminate the impact of higher-order oligomer contamination on the sample's elution peak. Samples in the presence of (GT)₁₉ or (GT)₂₄ were separated by using REGALS. In this program, the radius of gyration (Rg) and I(0) were calculated via Guinier fit. The molecular weight was estimated based on the correlation volume (Vc)[57]. Maximal extension Dmax was calculated via GNOM[58]. All these structural parameters (radius of gyration Rg [Å], maximal extension Dmax [Å], and molar mass MM [kDa]) shown in Supplementary Table 2 were acquired from averaged SAXS experimental curves, which were generated by calculating an average of several SAXS curves with a similar and stable Rg values allowing a better signal/noise ratio. Dimensionless Kratky plots were used to better visualized the flexibility/extendedness of different protein/oligonucleotides complex[59]. The accuracy of the fit is evaluated using χ² values derived from the two equations below (5) and (6)[60]:

$$\chi^2 = \frac{1}{N-1} \sum_j \left[ \frac{I_{\exp}(q_j) - cI_{\mathrm{theor}}(q_j)}{\sigma(q_j)} \right]^2 \tag{5}$$

where $N$ is the number of points in the experimental curve, $I_{exp}(q)$ is obtained by experimental data, $\sigma(q)$ are the experimental errors, $I_{theor}(q)$ is the calculated theoretical intensity and c is the scaling factor given as in Svergun et al. (1995)[61].

$$c = \left[ \sum_j \frac{I_{\exp}(q_j) I_{\mathrm{theor}}(q_j)}{\sigma(q_j)^2} \right] \bigg/ \left[ \sum_j \frac{I_{\mathrm{theor}}(q_j)^2}{\sigma(q_j)^2} \right] \tag{6}$$

## Construction of the 3D models of TDP-43 protein

3D models of wild type and R52E mutant TDP-43 protein (a.a., 1-277) have been generated with BIOVIA Discovery Studio Visualizer software (v24.1.0.23298, Dassault Systems, San Diego, 2023) using our model of TDP-43 RRM1-2 (a.a.,101-277) in complex with (GU)₁₂ previously published[24] and a TDP-43 NTD crystal structure (PDB, 5MDI). The unfolded linker region (a.a., 80-100) has been added between these two structures.

DADIMODO calculations were performed using the webserver (https://dadimodo.synchrotron-soleil.fr). The two RRM domains (residues from 100 to 277) and the RNA were defined as a unique rigid body. The two NTD domain (residues 1 to 77), from each TDP-43 (a.a., 1-277) unit, were defined as separated rigid body.

## Cross-linking Assay

100 pmol purified 1-277_WT or 1-277_R52E were incubated at 37 °C during 30 min in presence of 30 pmol of the oligonucleotide [GAA-GAAGAGAAGAAGAAG-A₆-(GT)₁₂-A₆-CTTCTTCTTCTCTTCTTC] named *Stem-loop (GT)₁₂* or 10 pmol of the oligonucleotide [GAAGAAGAGAA-GAAGAAG-(GT)₁₂-A₆-(GT)₁₂-A₆-(GT)₁₂-CTTCTTCTTCTCTTCTTC] named *Stem-loop (GT)₁₂A₆(GT)₁₂A₆(GT)₁₂*. The experiments were done in 20 mM Hepes buffer at pH 7.6, containing 25 mM KCl, 1 mM TCEP and 2 mM MgCl₂. 1 mM BS3 chemical arm (bis(sulfosuccinimidyl)suberate) was added into the incubated samples for cross-linking reaction. The reaction last 20 min at room temperature and then 100 mM Tris-HCl (pH 8.0) was added into the mixture to stop the reaction and remove the excess of BS3. The mixture was incubated for 20 min at room temperature. The cross-linked proteins were separated by SDS-PAGE (10% poly-acrylamide) and visualized by Coomassie-blue.

## Electrophoretic mobility shift assay (EMSA)

Previously purified TDP-43 fragments (a.a., 1-277) were incubated in the presence of 10 pmol of the oligonucleotides, stem-loop $(GT)_{36}$, stem-loop $A_{48}$ or stem-loop $A_{72}$ (Sequences in Supplementary Table 3). The stem structure facilitates the ethidium bromide visualization on a poly-acrylamide gel. The mixtures were incubated with 20 mM Hepes buffer at pH 7.6, containing 25 mM KCl, 1 mM TCEP, and 2 mM MgCl2 at room temperature for 20 min in final volume of 20 μL. Free and bound to protein oligonucleotides were separated in an 8 % poly-acrylamide gel in 0.5× TAE buffer at 80 V for 85 min on ice. Finally, gels were stained with 0.5 μg/mL of ethidium bromide.

## Cell culture and transfection

HeLa cells (obtained from ATCC, CCL-2) were cultured at 37 °C, 5% $CO_2$ in high glucose DMEM medium (Life Technologies) supplemented with 10% fetal bovine serum (FBS; Thermofisher), 100 units/mL of penicillin and 100 μg/mL of streptomycin (Life Technologies). Cells at confluence were plated in 96-well plates (PhenoPlate™, PerkinElmer) at a density of 18,000 cells per well and transfected with target plasmids (0.25 μg/well) or siRNA (0.2 μg/well) using Lipofectamine 2000 (Invitrogen) according to the manufacturer's protocol. Transfected cells were incubated for 24 h at 37 °C, 5% $CO_2$, and processed (treated or fixed) depending on the assay.

## Imaging parameters

There were no changes in the imaging parameters in the following assays. In addition, our pipelines from the cell detection to image analysis are entirely automated.

## Microtubule bench assay

HeLa cells were co-transfected with the plasmids encoding RFP-MBD- or GFP-MBD-fused proteins as described above. The indicated plasmids were obtained previously[25]. Mutations were introduced by site-directed mutagenesis and were validated by DNA sequencing (Eurofins Genomics). Following 24 h cells were fixed with ice-cold methanol for 10 min at -20 °C, washed once with PBS, and further fixed with 4% paraformaldehyde (PFA) diluted in PBS for 10 min at 37 °C. The cells were washed three times with PBS and stained with 4 μg/mL DAPI to visualize the nuclei.

The cell images were obtained with the Opera Phenix™ Plus High Content Screening System (PerkinElmer) in confocal mode using a ×40 NA 1.1 objective with water immersion.

Cell image analysis was performed using the Harmony v5.2 software (PerkinElmer) as described previously[28]. Briefly, we selected the population of co-transfected cells expressing both RFP- and GFP-fused proteins. Nuclei, cytoplasm, and microtubule clusters were detected automatically. Identified clusters of microtubules (spots) were then filtered by applying thresholds to remove non-tubular spots: spot area ($>20$ px$^2$ and $<300$ px$^2$) and spot width-to-length ratio ($<0.3$). The mean intensities of GFP and RFP in each selected microtubule cluster were measured and normalized to the mean fluorescence intensities in the cytoplasm. Mixing score (R-squared) was obtained by plotting the GFP vs RFP and applying a linear regression with automatic detection of $R^2$ values. Data analysis, including filtering and calculations, was performed using MATLAB R2023a.

## Nuclear aggregate assay

HeLa cells were transfected with the indicated plasmids encoding HA-fused proteins as described above. Plasmids harboring HA-TDP43 were obtained previously[25]. Mutations were introduced by site-directed mutagenesis and were validated by DNA sequencing (Eurofins Genomics). Following 24 h of transfection, cells were treated with the indicated concentration of arsenite or DMSO for 1 h at 37 °C. After treatment, cells were washed with PBS, and fixed with 4% PFA for 20 min at 37 °C. The cells were washed three times with PBS and incubated with the blocking solution (PBS, 2% BSA, 0.2% Triton X-100) for 1 h at room temperature. Next, cells were incubated with mouse monoclonal anti-HA antibodies F-7 (Santa Cruz Biotechnology, sc-7392, 1:1000) overnight at 4 °C with the following incubation with donkey anti-mouse DyLigh-488-conjugated secondary antibodies (Invitrogen, SA5-10166, 1:1000). In addition, rabbit polyclonal anti-G3BP1 antibodies (Sigma-Aldrich, G6046, 1:1000) and donkey anti-rabbit DyLigh-488-conjugated secondary antibodies (Invitrogen, SA5-10038, 1:1000) were used to detect G3BP1 and stress granules.

To visualize poly-A mRNA in HeLa cells, mRNA in situ hybridization was performed. Following the PFA fixation cells were incubated with ethanol 70% for 10 min at RT and with 1 M Tris-HCl pH 8.0 solution for 5 min. Next, cells were incubated with Cy3-conjugated oligo-dT probes (1 μg/mL, Molecular Probed Life Tech) in the hybridization buffer (0.005% BSA, 1 mg/mL yeast RNA, 10% dextran sulfate, 25% formamide in 2X SSC buffer). Finally, cell nuclei were stained using DAPI (4 μg/ mL).

The cell images were obtained with the Operetta CLS™ High-Content Analysis System (PerkinElmer) in confocal mode using a ×20 NA 1.0 objective with water immersion. Cell image analysis was performed using the Harmony v5.2 software (PerkinElmer). We selected the population of transfected cells expressing HA-fused proteins. HA-tag corresponds to a nona-peptide YPYDVPDYA derived from the hemagglutinin (HA) protein. This tag is used as antigen recognized by specific antibodies. Following nuclei detection using DAPI, granules of HA-tagged proteins in nuclear regions were detected automatically. The Harmony software calculated the relative intensity of nuclear aggregates as the ratio of (1) integrated signal from all pixels of all nuclear aggregates in a nucleus (intensity within the granule minus the local background intensity) and (2) integrated intensity over the nucleus. The enrichment of poly-A mRNA in nuclear aggregates was calculated as the ratio between the fluorescence of poly-A mRNA in a nuclear aggregate and the entire nucleus. Data analysis, including filtering and calculations, was performed using MATLAB R2023a.

## Splicing reporter assay

HeLa cells were transfected with the TDP43-specific siRNA (5'-CAC UAC AAU UGA UAU CAA A)TT-3', Eurofins Genomics) and non-silencing siRNA (AllStars Negative Control QIAGEN, 10272281) as described above. Cells were incubated for 24 h and transfected with the indicated plasmids encoding splicing reporter minigenes. Following 24 h of the transfection cells were washed with PBS, and fixed with 4% PFA for 20 min at 37 °C. The cells were washed three times with PBS, incubated with the blocking solution for 1 h at room temperature, and then incubated with anti-HA antibodies (1:1000, Santa Cruz Biotechnology) overnight at 4 °C. Next, cells were incubated with goat anti-mouse IRDye® 680RD secondary antibodies (LI-COR, 926-68070, 1:1000) for 1 h, at room temperature, washed with PBS, and incubated with DAPI for nuclei detection. Silencing efficiency was verified using rabbit anti-TDP43 antibodies (Proteintech, 12892-1-AP, 1:1000) and goat anti-rabbit IRDye® 680RD secondary antibodies (LI-COR, 926-68071, 1:1000).

The cell images were obtained with the Operetta CLS™ High-Content Analysis System (PerkinElmer) in confocal mode using a ×20 NA 0.4 objective. Cell image analysis was performed using the Harmony v5.2 software (PerkinElmer). Following nuclei detection using DAPI, fluorescent intensities of GFP and RFP in the nuclear regions were measured.

## Reporting summary

Further information on research design is available in the Nature Portfolio Reporting Summary linked to this article.

## Data availability

All data generated or analysed during this study are included in the manuscript and supporting files. The SAXS data has been deposited on the SAXBDB database with the accession code SASDXL8. BMRB codes of previously published assignments used in this study are 30345, 27613 and 19290. The PDB code of a previously published structure used in this study is 5MDI. Source Data is provided as a Source Data file. Source data are provided with this paper.

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

## Acknowledgements

This work was supported by INSERM, CNRS, Université Evry Paris-Saclay (FRR grants), Université Paris-Saclay, and Genopole (SATURNE grant). This work was also supported by Agence Nationale de la Recherche (ANR) (Grant: ANR-24-CE44-3867-01-SUFATOP). Y.F. was the recipient of a PhD scholarship from Doctoral School SDSV (ED n°577) and this work is part of Y.F.'s PhD thesis project at Université Evry Paris-Saclay/Université Paris-Saclay, conducted toward the Doctor of biochemistry and structural biology degree. We acknowledge SOLEIL (Saint-Aubin, France) for provision of synchrotron radiation facilities and we would like to thank all the staff for assistance in using beamline SWING under proposals 20220453 and 20230118. Financial support from the IR INFRANALYTICS FR2054 for conducting the research is gratefully acknowledged.

## Author contributions

Y.F., performed the experiments, data analysis, validation, preparation of the figures and discussion of the results. Y.F. and M.-J. C., NMR investigation, analysis and validation; Y.F. and J.C R-G, protein purification; Y.F. and A.T., SAXS investigation, analysis and validation; V.J. and S.P., cellular investigation, analysis and validation; D.P. and A.B., conceptualization, formal analysis, investigation, validation, writing–review and editing; A.B., Project management and funding acquisition.

## Competing interests

The authors declare no competing interests.
