## [Transparent Peer Review file · Nature Communications]

From TDP-43/RNA complex formation to disease-linked TDP-43 aggregation through a structural and cellular approach

Corresponding Author: Dr Ahmed Bouhss

Version 0:

Reviewer comments:

Reviewer #1

(Remarks to the Author)

The role of RNA-binding proteins (RBP) in RNA metabolism is currently subject to numerous investigations with many questions yet to be answered. Feng et al., carried out a detailed structural analysis of the RBP, TDP-43, using Nuclear Magnetic Resonance, Isothermal Titration Calorimetry, Small Angle X-Ray Scattering, and cellular experiments. They indicate the vital role nuclear interaction of intronic GU-rich RNA with TDP-43 N-terminal domain and RNA-recognition motifs, plays in preventing TDP-43 aggregation, and important process in the pathology of neurodegeneration. Oxidative stress of cells upon exposure to arsenite induced TDP-43 aggregation modulated by GU-rich RNA. Finally, the authors work suggested that stress conditions induced acetylation of TDP-43 in some RRM residues, leading to TDP-43 aggregation. The work is interesting and builds off similar findings from other groups in this area over the last ~10 years, although there is some over-interpretation of the cellular data which requires further biochemical analysis to confirm findings.

Major Points

- Introduction Line 60 – which part of TDP-43 the authors are referring to as the NTD is unclear. Stated ‘no RBP other than TDP-43 itself harbors a similar NTD domain’, this is true, although other RBPs have similar RRM (albeit not within their NTDs, eg FUS has similar RRM in its CTD). These descriptions should be more clear – it appears the authors refer to the NTD as the sequence N-terminal to RRM1 in later parts of the paper but this is vague in Introduction. NTD and RRM acronyms should be defined in abstract.
- Fig 1A is not to scale with a comparatively shortened CTD shown; this should be clarified.
- TDP-43 clearly preferentially binds to GU-rich RNA. The authors could consider showing control data (separate from the polyA sequence already provided) indicating how CA-rich RNA (usually used as control for GU RNA in previous studies) influences TDP-43 RRM1=RRM2 interaction and interactions between the NTDs of TDP-43 monomers.
- The authors use either the E17R or R52E mutant to prevent intermolecular association between TDP-43 NTD. However previous reports have used 6-point mutations in the NTD to prevent TDP-43 oligomerisation – 6M (E14, E17, E21, Q34, R52 and R55). The authors should show data that the single R52 mutant is sufficient to completely prevent TDP-43 interaction and how this compares to 6M mutant to allow for comparison to previous studies. Otherwise, the authors should include some 6M data in their RNA NMR experiments, or discuss differences with the previous literature.
- The SAXS data observation of compacted nucleoprotein complex is only observed in the truncated TDP-43 NTD/RRM (Fig. 3). The authors should conjecture on how this is translated to full length TDP-43 compaction in these RBP complexes.
- Data presentation is sometimes confusing – eg Fig 4C,D,E y axes start at non-zero which over-conflates the changes seen, and arrows drawn on graphs are not needed. %change on graph is not informative, instead should report on statistical analyses. Also, the 8 datapoints appear to be technical rather than experimental replicates (stated that each is a separate well) – these data should be analysed with biological replicates from separate experiments conducted independently.
- Fig 5 shows effects of arsenite on TDP-43 aggregation, which is usually taken as a biochemically determined feature – to confirm ‘aggregation’, authors should perform experiments to determine whether there are differential effects on soluble/insoluble protein ratio of mutants with arsenite treatment. This is particularly important given the focus on ‘soluble proteins’ (the data presented does not in fact investigate solubility), and also as there appears to be some difference in image quality (eg HA-FUS in Fig 5B is clearly not the same resolution/focus as HA-TDP-43) which could account for the lack of detection of puncta in response to stress. Were all images collected using the same parameters? This issue is also

somewhat confounded in the text with interchangeable use of the term 'granule' in the figures/methods and 'aggregate' in the text (eg Supp Fig 14 – are the authors referring to 2 different structures or are they the same, and what are the definitions of granule vs aggregate?) – closer attention to consistency would assist the reader interpret findings. Without biochemical confirmation, the focus on solubility and aggregation is weak.

- The authors could further explore how GU-rich RNA TDP-43 interactions intersects with TDP-43 acetylation that has been detected in TDP-43 pathology related to aggregation to provide even further disease context by using the K145Q and K192Q acetylation-mimic mutants that recapitulate changes seen in ALS spinal cord pathology (in addition to the data on K-R acetylation blocking mutants).

Reviewer #2

(Remarks to the Author)

The manuscript by Feng et al. provides important mechanistic insights into how TDP-43, a key RNA-binding protein implicated in ALS and FTL, forms both physiological and pathological assemblies. Extending their previous work, the authors systematically examine the interplay between TDP-43's N-terminal domain (NTD)-mediated self-association and its cooperative binding to GU-rich RNA, employing a combination of biochemical, structural (NMR, SAXS), and thermodynamic (ITC) approaches, along with cellular assays.

Their premise is that under normal nuclear conditions, TDP-43 binds cooperatively along GU-rich intronic RNA, suppressing aberrant NTD:NTD interactions while promoting RNA compaction through long-range clustering of TDP-43:RNA complexes. In contrast, under oxidative stress or in the cytoplasm, where GU-rich RNAs are limited, disruption of cooperative RNA binding permits pathological NTD:NTD interactions that drive TDP-43 aggregation. Moreover, they show that expression of GU-rich RNA can protect against stress-induced aggregation.

This work highlights the dual role of TDP-43's NTD and RNA-binding domains in both functional RNA processing and pathological aggregation. The authors present an impressively large amount of high-quality, carefully controlled data, which they meticulously analyze to support their model. However, several questions regarding their interpretation need to be addressed before the manuscript is suitable for publication.

Major Points:

1. The NMR line-broadening data supporting the hypothesis that cooperative binding of the RRM1/2 domains to GT12 DNA prevents NTD interactions are not as convincing as claimed. The authors focus primarily on residues that support their model (e.g., Figs. 2, S5, S6, S8). What about other key interface residues such as R52, E17 (a particularly critical residue), R55, G53, and T30? For example, in Fig. S8, while T32 and E21 show increased intensity in GT12-bound samples, I18 and I16 do not. Additionally, the molecular size differences between GT6- and GT12-bound samples complicate direct intensity comparisons (e.g., E156, which likely does not engage in NTD interactions, varies between conditions). Some affected peaks remain unlabeled, such as a Trp indole NH and a peak near 6.3 ppm.
2. Even if the authors can convincingly demonstrate that peak intensities are partially recovered in the presence of GT12, the interface residue signals do not fully return to levels observed in the absence of NTD interactions, suggesting that significant NTD interactions persist when bound to GT12 (Fig. 2B). The manuscript overemphasizes these findings, and together with the SAXS data, seems to suggest that cooperative RNA binding completely eliminates NTD:NTD interactions, which is not fully supported by the data.
3. It remains unclear why the NTD would be unable to engage in intermolecular interactions when TDP-43 is complexed with DNA/RNA (GT6 or GT12). The authors provide evidence that the NTD does not directly bind RNA or the RRM1/2 domains and that the RNA:TDP-43 complexes are monodisperse by SEC. What, then, prevents NTD self-association under these conditions? Alternatively, are NTD:NTD interactions simply transient, only becoming significant at higher local concentrations, such as when multiple TDP-43 protomers are bound to RNA?
4. In the SAXS model presented in Fig. S11, the two NTDs appear close enough to interact. Additional analysis such as considering linker lengths, distances between RRM1/2 domains, and RRM dimensions would strengthen the argument that NTDs remain separated. Furthermore, the SAXS data suggest the presence of multiple conformations, making it potentially inappropriate to present a single "best-fit" model as definitive.
5. Analyzing the populations of bound and unbound states would better demonstrate whether, in the cooperatively bound samples, NTDs are largely unassociated. Such analysis could also help resolve concerns mentioned in the points above. Additionally, the authors should discuss (or quantify) the dynamic nature of NTD interactions in detail, as their own NMR data (e.g., peak splitting in Figs. 2, 6, and 8) suggest the presence of multiple conformations.
6. Statements suggesting that "cooperative binding of TDP-43 to long (GT12) repeats weakens interactions between NTDs" (e.g., p.6, line 254) are potentially misleading. These phrases imply direct regulation of NTD association by RNA binding, yet the data do not clearly support a direct mechanistic link. Such statements should be revised for clarity and accuracy.
7. The model shown in Fig. 7B is misleading, as it does not clearly depict why NTDs from adjacent TDP-43 protomers cannot interact. The figure should be revised for clarity.

Minor Points:

8. Line 374 – "act together" should be replaced with "each contribute."

9. Some sections of the manuscript are repetitive and could be streamlined to improve readability.
10. The physical meaning of negative chemical shift perturbations (CSPs) in Fig. 1 and S1 is unclear. Correlation plots comparing CSPs between conditions may be more effective in illustrating differences and avoiding confusion.
11. The main text figures should include error estimates for ITC-derived KD values.
12. The equations used for fitting the ITC and SAXS data should be included in the Methods section.
13. Figures are referenced out of order in some sections (e.g., Fig. S17 etc.).
14. Some reported fitting errors for the ITC data (Table S2) seem small and should be double-checked.
15. The manuscript would benefit from thorough editing to correct pluralization errors and missing articles throughout the text.

Reviewer #3

(Remarks to the Author)

The binding studies are particularly interesting and provide new mechanistic insights into TDP-43 assembly. The results are clearly presented, supported by multiple complementary methods, and I highly recommend this manuscript for publication.

Some minor issues

1) SAXS-related issues

1.1 Please deposit (at least WT SAXS data) in the SASBDB.

1.2 In general consider showing $p(r)$ functions when discussing the changes in size. Kratky plots are shown throughout the manuscript but the information gained from the plot is not stated.

Eg. in line 277: Reference to SupplementFigure 9B is misleading at this point -- this is Kratky Plot and does not refer to D_{max} . Here comparison of $P(r)$ functions would emphasize this point (misleading figure legend in Sup FS9B, E17R+R52E, but not E17R alone, reduced the sizes of the complexes in the presence of (GT)₆A₁₂(GT)₆.

1.3 In general, the SAXS results could be strengthened by including quantitative comparisons—for example, reporting the D_{max} : for wild-type TDP-43 (162 nm) alongside the range observed for the mutants (159–165 nm).  and quantify accordingly in line 286 as well the increase in D_{max} values (line 297) and later in text. Is there a way to estimate the errors from D_{max} ? (Obviously the information is given in various tables, but could help the reader).

1.4 Elution profiles

Supplementary Figure S9C -- Please comment on the shoulders - such as for the E17R mutants

Supplementary Figure 11 -- Comment on the shoulders at frame 100 -- are these aggregates - higher oligomeric structures?

Minor (SAXS section):

- line 265
- The chromatograms and the SAXS images were recorded during the elution. We observed that the elution volumes were similar for all the conditions in the presence of (GT)₁₂ for which the binding is cooperative."
- To improve clarity, consider explaining the underlying observations in a bit more detail eg. The single mutants (blue and yellow trace) elute similarly as WT (red trace), which shows cooperative binding (see ITC Fig 1D, red plot)
- Supplementary Figure S9A -- I'm not sure what the difference is between the upper and lower panels. Perhaps a copy/paste issue with the legend?
- line 260 -- Restructure: To investigate structural changes on a larger scale than accessible by NMR, we performed SAXS analysis of in vitro-purified TDP-43-nucleic acid complexes.
- line 264: The acquisition of the SAXS images WAS..... or SAXS measurements were performed in-line with size-exclusion chromatography (SEC)
- line 265 -- specify which chromatogram (UV 280nm + 260nm ?)
- line 273 : Already here, place the reference to Figure 3 and SAXS Table in supplementary information. (see line 277)
- line 274: As two hypotheses were formulated earlier be specific which one is in agreement.
- Line 1940 -- in Methods the use of REGALS for decomposing overlapping peaks is mentioned -- was this observed in the measurements?
- Line 1945 -- not clear what is meant with: were acquired from averaged SAXS experimental curves.
- Line 1950 -- models ... HAVE been generated

General:

- To improve clarity and conciseness, consider introducing the E52R and R17E mutants, along with the different nucleotide constructs such as (GT)₆A₁₂(GT)₆, earlier in the text. This would reduce the need for repeated reintroductions later in the manuscript.
- line 277m, replace "Accordingly" with 'next', 'in addition', ...
- line 19 -- not clear which mechanisms are being referred to
- line 29 -- define RRM at this point
- line 30 -- is it assembly of TDP-43 in (OR ON ?) introns

- line 144 -- better: .. and includes the full NTD domain. (instead of while keeping)
- line 148 -- ..we compared the chemical shifts of the RRM1-2 residues to constucts containing (1-277) and lacking (101-277) the NTD domain
- line 167 -- replace whatever with whether
- line 170 -- In THE presence of...
- line 172 -- remove 'a' to read: a major feature of cooperative association with (GT)₁₂
- line 177 -- perhaps add 'red plot (Figure 1D)
- line 178 -- why not mention the second mutant?
- line 183 -- add "green plot"
- line 214 -- 'A' similar result...
- line 267 -- add: the binding is cooperative (WT AND SINGLE MUTANTS).
- line 290 -- remove 'The' before 'similar results'
- line 317 -- replace 'To document' with to assess or investigate or characterise...
- line 553 -- mutated TDP/43*aa. 1/277
- line 382 -- define 'HA'
- line 401 -- use passive form : Two pairs of lysine residues, known to undergo acetylation following arsenite stress, were mutated to arginine.
- Supplementary Figure 2B: One could first show R52E mutant then the double mutant
- Supplementary Figure 3: Color names should be in lowercase rather than uppercase.
- line 1089 -- interactions DO not interfere
- line 1098 -- Significant CSPs ARE OBSERVED only in the binding residues of RRM1-2 (assigned in black).
- line 1099 -- COMPARED to
- line 1142 -- Zoomed-in VIEW on THE chemical shift perturbations FOR TDP-43 residues in the RRM, with or without the indicated nucleotidess
- line 1144 -- R52E not R525E
- line 1927 -- Data collection parameters ARE shown
- line 1934 -- In each experiment -- (remove 's')

Reviewer #4

(Remarks to the Author)

Version 1:

Reviewer comments:

Reviewer #2

(Remarks to the Author)

I appreciate the authors efforts and their responsiveness to my concerns. I beleive all major concerns have been addressed and the manuscript is much improved.

One small remaining concern, I had asked that the authors include the equations they used to fit their data, and while they did this with the SAXS analysis, they provided a link to the ITC instrument's manual. I note that this document is on a password-protected site requiring registration to access. Further, since this is a commercial site, the links and document may be updated/revised/moved/discontinued etc thus not preserving critical information needed to reproduce the results of this paper. For full transparency and completeness, the authors should include the equations used to analyze the ITC data in the material and methods section. If length is a concern, then can be included in an expanded M&M section ion supplemental material.

Reviewer #3

(Remarks to the Author)

All my comments have been addressed, and several figures have been added. In addition, some text passages have been extended for greater clarity.

I recommend that the manuscript be accepted for publication.

Reviewer 1:

Q1: Introduction Line 60 – which part of TDP-43 the authors are referring to as the NTD is unclear. Stated ‘no RBP other than TDP-43 itself harbors a similar NTD domain’, this is true, although other RBPs have similar RRM1s (albeit not within their NTDs, e.g. FUS has similar RRM1s in its CTD). These descriptions should be more clear – it appears the authors refer to the NTD as the sequence N-terminal to RRM1 in later parts of the paper but this is vague in Introduction. NTD and RRM1 acronyms should be defined in abstract.

We agree with the reviewer. We indicated the meaning of the acronyms in the abstract. In addition, we added a sentence in the text to clarify the position of the NTD domain (Page 2, lines 59-61).

Q2: Fig 1A is not to scale with a comparatively shortened CTD shown; this should be clarified.

Thank you for noticing this. Figure 1A has been corrected.

Q3: TDP-43 clearly preferentially binds to GU-rich RNA. The authors could consider showing control data (separate from the polyA sequence already provided) indicating how CA-rich RNA (usually used as control for GU RNA in previous studies) influences TDP-43 RRM1=RRM2 interaction and interactions between the NTDs of TDP-43 monomers.

We agree with the reviewer. In the revised manuscript, we provided new data to answer this point. In the Supplementary Figure 5, new EMSA results show that wild-type or mutant forms do not bind to A₄₈ nor A₇₂ compared to (GT)₃₆ oligonucleotide. Moreover, NMR results show no significant shift observed in spectra of wild type or mutant in presence A₁₂ or (CA)₆ oligonucleotides, comparing to protein alone. Therefore, TDP-43 poorly interacts with poly A and CA-rich oligonucleotides compared to GT repeats.

A comment has been added to text to present these additional controls (Page 5, lines 183-186).

Q4: The authors use either the E17R or R52E mutant to prevent intermolecular association between TDP-43 NTD. However, previous reports have used 6-point mutations in the NTD to prevent TDP-43 oligomerisation – 6M (E14, E17, E21, Q34, R52 and R55). The authors should show data that the single R52 mutant is sufficient to completely prevent TDP-43 interaction and how this compares to 6M mutant to allow for comparison to previous studies. Otherwise, the authors should include some 6M data in their RNA NMR experiments, or discuss differences with the previous literature.

We understand the point raised by the reviewer. However, in our hand, a single mutation, R52E or E17R, in the NTD was enough to significantly disrupt NTD/NTD interactions. Indeed, the elution of 1-277_WT, 1-277_E17R and 1-277_R52E by a Size Exclusion Chromatography (SEC) column was shown in the new Supplementary Figure 1. Mutant E17R and mutant R52E display lower elution volumes than wild type TDP-43 due to the disruption of NTD/NTD interactions. The molecular weight (MW) of R52E mutant, E17R mutant and WT predicted by SAXS experiments are 34.5kDa, 34kDa and 44.1kDa, respectively. The predicted MW of R52E and E17R is much closer to the theoretical value (MW 31.5kDa) than WT. Moreover, we observed larger aggregates during the elution of wild type TDP-43 than two single mutants.

Q5: The SAXS data observation of compacted nucleoprotein complex is only observed in the truncated TDP-43 NTD/RRM (Fig. 3). The authors should conjecture on how this is translated to full length TDP-43 compaction in these RBP complexes.

Indeed, we used truncated TDP-43 *in vitro* because the unstructured CTD leads to TDP-43 aggregation *in vitro*. TDP-43 CTD may also contribute to the compaction of nucleoprotein complex through additional homotypic interactions. In addition, TDP-43 CTD may also enable heterotypic interactions with protein partners in the nucleus. A comment has been added in the discussion (Page 12, lines 508-513).

Q6: Data presentation is sometimes confusing as – eg Fig 4C,D,E y axes start at non-zero which over-conflates the changes seen, and arrows drawn on graphs are not needed. %change on graph is not informative, instead should report on statistical analyses. Also, the 8 datapoints appear to be technical rather than experimental replicates (stated that each is a separate well) – these data should be analyzed with biological replicates from separate experiments conducted independently.

We agree that the scales shown in Figure 4 do not start at zero. We did this to facilitate the reading and visualize the variations. This is why we would like to ask the reviewer if we could keep this representation of the data. The values shown correspond to the R^2 value. For all the variations mentioned in the text, we verified their significance through statistical analysis. Regarding the experimental replicates, we now provide at least one replicate obtained on independent plates and performed on a different week. The analyses for experimental replicates have been added in the Supplementary Figure 20.

Q7: Fig 5 shows effects of arsenite on TDP-43 aggregation, which is usually taken as a biochemically determined feature – to confirm ‘aggregation’, authors should perform experiments to determine whether there are differential effects on soluble/insoluble protein ratio

of mutants with arsenite treatment. This is particularly important given the focus on ‘soluble proteins’ (the data presented does not in fact investigate solubility), and also as there appears to be some difference in image quality (eg HA-FUS in Fig 5B is clearly not the same resolution/focus as HA-TDP-43) which could account for the lack of detection of puncta in response to stress. Were all images collected using the same parameters? This issue is also somewhat confounded in the text with interchangeable use of the term ‘granule’ in the figures/methods and ‘aggregate’ in the text (eg Supp Fig 14 – are the authors referring to 2 different structures or are they the same, and what are the definitions of granule vs aggregate?) – closer attention to consistency would assist the reader interpret findings. Without biochemical confirmation, the focus on solubility and aggregation is weak.

TDP-43 aggregation after RRM acetylation upon arsenite exposure in cells has been already reported in the literature (Ref 26, Cohen et al, Nature Com, 2015).

We agree about the point raised by the reviewer to provide additional controls. We performed two experiments to explore the reversibility of TDP-43 assemblies in the nucleus after arsenite treatments. First, we considered whether nuclear TDP-43 could be extracted through detergent treatment to briefly permeabilize the cell membrane after arsenite treatments. In the new Supplementary Figure 22, after washing with 0.1% Triton x100 for 5 min, we found nuclear TDP-43 levels were little affected for WT and G146A, but decreased for mutants R52E-G146A and R52E. This is in agreement with a higher solubility of R52E-G146A and mutant R52E than WT and G146A upon arsenite treatment.

Second, in the new Supplementary Figure 23, we tested the reversibility of nuclear TDP-43 aggregates/granules formed after arsenite treatments. To this end, after arsenite treatments, arsenite was washed out for 90 min. As expected, we found that G3BP1 stress granules (here 200 μ M of arsenite was used to trigger a robust stress granule assembly) dissociate completely in agreement with the reversible nature of these RBP-rich condensates. In contrast, aggregates/granules of TDP-43_WT and TDP-43_G146A formed with 80 μ M of arsenite are still there, even though we evidenced a partial dissociation (which may be due to the partial removal of aggregates by cells during 90 min). Therefore, we considered TDP-43 aggregates/granules after arsenite treatment as aggregates in the text (though they may as well be solid-like condensates of very low kinetics). This result is in line with the previous knowledge of TDP-43 aggregation upon arsenite treatments and with the absence of mRNA enrichment in these assemblies (Figure 5A).

Regarding the suggestion to perform biochemical analyses, we think this hardly possible with TDP-43 since it has a strong tendency for aggregation *in vitro* after cell lysis. Therefore, we think that cell assays are more informative concerning this point than biochemical assays.

In response to Q7, two new Supplementary Figures 22 and 23 have been added to manuscripts and their results commented in the text and supplementary figure legends (Page 10, lines 419-427).

Q8: The authors could further explore how GU-rich RNA TDP-43 interactions intersects with TDP-43 acetylation that has been detected in TDP-43 pathology related to aggregation to provide even further disease context by using the K145Q and K192Q acetylation-mimic mutants that recapitulate changes seen in ALS spinal cord pathology (in addition to the data on K-R acetylation blocking mutants).

This is an interesting point. We generated the K145Q and K192Q acetylation-mimic mutants. The results have been added in the Supplementary Figure 24A. After arsenite treatment, Mutants K145Q-K192Q or G146A-K145Q-K192Q form nuclear aggregates. In contrast, nuclear aggregation of TDP-43 in cells expressing R52E-K145Q-K192Q or R52E-G146A-K145Q-K192Q is limited. Therefore, the acetylation of RRM lysine residues contributes to TDP-43 aggregation. A comment has been added in the text (Page 10, lines 429-436).

Reviewer 2:

Q1: The NMR line-broadening data supporting the hypothesis that cooperative binding of the RRM to GT12 DNA prevents NTD interactions are not as convincing as claimed. The authors focus primarily on residues that support their model (e.g., Figs. 2, S5, S6, S8). What about other key interface residues such as R52, E17 (a particularly critical residue), R55, G53, and T30? For example, in Fig. S8, while T32 and E21 show increased intensity in GT12-bound samples, I18 and I16 do not. Additionally, the molecular size differences between GT6- and GT12-bound samples complicate direct intensity comparisons (e.g., E156, which likely does not engage in NTD interactions, varies between conditions). Some affected peaks remain unlabeled, such as a Trp indole NH and a peak near 6.3 ppm.

We understand the point raised by the reviewer. In Figure 2A, we analyzed and measured the variations in CSPs when R52E and wild-type TDP-43 interacts with (GT)₆ or (GT)₁₂. For these experiments, we agree that the analyzes of the peak heights are more complicated to analyze than CSPs because of the difference of molecular weight in the presence of (GT)₁₂ versus (GT)₆, which may explain why some residues such as I18 and I16 do not show increased intensity in the presence of (GT)₁₂.

To answer the reviewer's point, we now show and analyze the CSPs of some interface residues, including E17, R55, G53, that were missing as suggested by the reviewer (Supplementary Figure 11 and Figure 2A). These residues show more significant chemical shifts when WT (a.a. 1-277) binds to (GT)₁₂ than when WT binds to (GT)₆. In contrast, the same residues do not show any chemical shifts in any condition for the mutant R52E. This is in line with our hypothesis. A sentence has also been added to the text (Page 6, lines 226-229). R52E was not shown because the analysis is based on comparing the CSPs between wild-type TDP-43 and R52E for which R52 is mutated into E. T30 CSPs cannot be observed because of the superimposed presence of additional peaks such as that of R171.

In addition, let us note that we performed competition assays with E17R (Figure 2D). Here we show the relative variations in peak height for all the NTD residues of TDP-43 R52E mutations. The normalization of the variations of the peak heights allows to withdraw the bias due to the molecular weight differences when TDP-43 interact with (GT)₆A₁₂(GT)₆ or (GT)₁₂ (Figure 2D). The analysis of the ratio of peak heights in Figure 2D, indicate that I18, I16, I21 but also S29, V31 and T32 showed an increase in peak height in (GT)₁₂-bound samples compared to (GT)₆A₁₂(GT)₆ when E17R NTD was added. These residues are located just in front of R52 in the NTD dimer (Figure 2C), so that these results make sense. Residues that do not participate in this interface do not show a relative increase in peak heights. It is also the case for the residues that are located in the opposite interface around R52E. Let us remind that this R52E-related interface is no longer functional because the mutated interface (E52-related interface) in the R52E mutant (¹⁵N-labelled TDP-43, a.a. 1-277) cannot interact with the mutated interface (R17-related interface) in E17R NTD (unlabeled NTD). Indeed, these respective substitutions

(changing both the charge and the size of initial residue) hinders the constitution of the interface. Only the E17-related interface in the R52E mutant can interact with the R52-related interface in E17R NTD.

Regarding W68, it is not directly involved in NTD/NTD interactions (Wang et al, EMBO, 2018). Accordingly, we did not observe any relative increase in peak height ratio for the W68 backbone (Figure 2D). However, indeed, as noticed by the reviewer, there is an unexpected increase in CSP of this residue in wild type TDP-43 in the presence of (GT)₁₂ in comparison with (GT)₆ (now indicated in Supplementary Figure 11). Since W68 is located in a flexible domain, the side chain may be sensitive to change in local structural environment. Regarding E156, E156 is located in a different domain (RRM1) far from the NTD interface in RRM1, which may be as well sensitive to change in local environment. We now indicated that we focus our attention on NTD residues for the results presented in Figure 2 (Page 5, lines 213-215). The reviewer also mentioned a peak near 6.3 ppm. In Fig. S8 (now Supplementary Figure 12), in the presence of oligonucleotide, the peak near 6.3 ppm corresponds to F229, which is not located in the NTD but in RRM2.

Q2: Even if the authors can convincingly demonstrate that peak intensities are partially recovered in the presence of GT12, the interface residue signals do not fully return to levels observed in the absence of NTD interactions, suggesting that significant NTD interactions persist when bound to GT12 (Fig. 2B). The manuscript overemphasizes these findings, and together with the SAXS data, seems to suggest that cooperative RNA binding completely eliminates NTD:NTD interactions, which is not fully supported by the data.

We agree with the reviewer that the peaks do not recover their initial height, which seems to indicate that interactions between NTDs still take place. We tuned down our conclusion and indicated that TDP-43 cooperative assembly in the presence of (GT)₁₂/(GU)₁₂ reduces NTD/NTD interactions instead of totally suppressing them.

However, the signals of NTD residues detected by NMR represent an average value of several dynamic interactions over time. For example, in the presence of (GT)₁₂, one TDP-43 protein can transiently be released from RNA/DNA to initiate NTD/NTD interactions with an adjacent TDP-43. Therefore, even if the peak intensities of NTD residues do not fully return to their initial levels corresponding to the absence of NTD/NTD interactions, we cannot exclude nor affirm that TDP-43 cooperative binding totally prevent NTD interactions after the analysis of this experiment. We can just indicate that cooperative TDP-43 association to RNA/DNA limits NTD/NTD interactions.

Q3: It remains unclear why the NTD would be unable to engage in intermolecular interactions when TDP-43 is complexed with DNA/RNA (GT6 or GT12). The authors provide evidence that

the NTD does not directly bind RNA or the RRM1s and that the RNA:TDP-43 complexes are monodisperse by SEC. What, then, prevents NTD self-association under these conditions? Alternatively, are NTD:NTD interactions simply transient, only becoming significant at higher local concentrations, such as when multiple TDP-43 protomers are bound to RNA?

According to our model, the separation distance between the NTDs of adjacent TDP-43 units is large (Supplementary Figure 16, see text in Page 7, lines 285-288). Their interaction is therefore possible, as indicated by the reviewer, but it would occur at the expense of an entropy penalty.

We agree with the reviewer. NTD/NTD are indeed transient and the occurrence of NTD/NTD interactions increases at high concentration (Ref. 18, Wang et al, EMBO, 2018). If a single TDP-43 monomer is bound to (GT)₆, the local concentration of NTD does not increase in the test tube. However, in the presence of (GT)₁₂/(GU)₁₂, when two TDP-43 are associated with RNA/DNA, we may expect that the chances of having NTD/NTD interactions should increase. However, the point is that we observed that the cooperative association of TDP-43 to (GT)₁₂/(GU)₁₂ negatively affects NTD/NTD interactions (Figure 2D) and *vice versa* (Figure 1D).

However, in agreement with the point of the reviewer, multiple TDP-43 bound to mRNA may provide an additional contribution to explain why the cooperativity-deficient mutant, G146A, is particularly prone to aggregation. Indeed, the binding of multiple G146A to long mRNA is possible which should promote the chances of NTD/NTD interactions between nearby TDP-43 in the absence of cooperativity (free from the structural constraints imposed by the cooperativity).

Q4: In the SAXS model presented in Fig. S11, the two NTDs appear close enough to interact. Additional analysis such as considering linker lengths, distances between RRM1/2 domains, and RRM dimensions would strengthen the argument that NTDs remain separated. Furthermore, the SAXS data suggest the presence of multiple conformations, making it potentially inappropriate to present a single “best-fit” model as definitive.

We agree with the reviewer. They could interact but the unstructured linker between NTD and RRM1 will be stretched, which may lead to an entropy penalty. All the calculations performed using DADIMODO resulted in good fits to the SAXS data, with χ^2 values around 1.3–1.4. In all the unique solutions obtained, the NTD domains were positioned far apart, with distances ranging from 80 to 100 Å. The length of the linker between the RRM and NTD domains was fixed at 12 residues (residues 94 to 105). The superimposition of the five best results has been added to the Supplementary Figure 16.

We agree that more than a single “best model” should be showed. The DADIMODO software indeed provides a single representative model that best fits the SAXS curve. These results indicate that a unique model with the two NTDs interacting directly is not compatible with the SAXS data. However, SAXS cannot fully exclude the presence of a minor population of

conformations in solution where the NTDs are close. This point is detailed below in the response to question 5.

A comment has been added regarding this point (Page 7, lines 289-294).

Q5: Analyzing the populations of bound and unbound states would better demonstrate whether, in the cooperatively bound samples, NTDs are largely unassociated. Such analysis could also help resolve concerns mentioned in the points above. Additionally, the authors should discuss (or quantify) the dynamic nature of NTD interactions in detail, as their own NMR data (e.g., peak splitting in Figs. 2, 6, and 8) suggest the presence of multiple conformations.

We agree with the reviewer and generated two closed state models displaying NTD/NTD interface between adjacent TDP-43 along (GU)₁₂, with distance constraints (4-5 Å) between R52 residue from NTD of one TDP-43 unit, and E17 residue from NTD of the second TDP-43 unit, and compared them to the experimental curve (Supplementary Figure 17).

The best-fitting models are linked to significantly elevated χ^2 values, 3.12 or 3.60, when R52 (from the first unit) interacts with E17 (from the second unit) or vice versa (E17/R52), respectively, which are obviously worse than the value from the open state model ($\chi^2= 1.35$). Then, the best-constrained model ($\chi^2= 3.12$) was subsequently used as the starting point for a new calculation performed without any distance constraints by using DADIMODO program. According to the new calculations, the best conformation was obtained, with a χ^2 value of 1.33 similar to the initial calculations (Supplementary Figure 16), corresponds to an open model in which the two NTD domains have moved apart (have separated) by approximately 72Å. We would like to note that all of the best-generated solutions displayed the models where the two adjacent NTDs are separated. This analysis has been added in text (Page 7, lines 289-302). We thank the reviewer for his comment.

In summary, even if NMR and SAXS experiments are performed under different conditions, the analysis of these data indicate that the cooperative association of TDP-43 to RNA limits NTD/NTD interactions.

Q6: Statements suggesting that "cooperative binding of TDP-43 to long (GT)₁₂ repeats weakens interactions between NTDs" (e.g., p.6, line 254) are potentially misleading. These phrases imply direct regulation of NTD association by RNA binding, yet the data do not clearly support a direct mechanistic link. Such statements should be revised for clarity and accuracy.

We agree with the reviewer. The sentence has been rephrased: "NTD/NTD interactions between adjacent TDP-43 are limited when TDP-43 binds cooperatively to long (GT)₁₂ repeats." (Page 6, lines 255-256). Many statements have been changed in the text to clarify this point.

Q7: The model shown in Fig. 7B is misleading, as it does not clearly depict why NTDs from adjacent TDP-43 protomers cannot interact. The figure should be revised for clarity.

We agree with reviewer. Figure 7b has been revised. We indicate the large separation distance between NTD of adjacent TDP-43 limits NTD/NTD interactions.

Q8: Line 374 – "act together" should be replaced with "each contribute."

We agree with reviewer. It has been changed (Page 9, line 393).

Q9: Some sections of the manuscript are repetitive and could be streamlined to improve readability.

We thank to the reviewer. We tried to improve the clarity of the manuscript to the best we can.

Q10: The physical meaning of negative chemical shift perturbations (CSPs) in Fig. 1 and S1 is unclear. Correlation plots comparing CSPs between conditions may be more effective in illustrating differences and avoiding confusion.

We agree with the reviewer. In the revised manuscript, we compared CSPs related to RRM1-2 between R52E and E17R+R52E (Figure 1B), R52E and WT (Supplementary Figure 2B). No significant changes for most of the residues in RRM1-2, so NTD/NTD interactions do not affect RRM domains.

Q11: The main text figures should include error estimates for ITC-derived K_D values.

We agree with the reviewer. Error estimates for ITC-derived K_D values are now shown in Figure 1C and D.

Q12: The equations used for fitting the ITC and SAXS data should be included in the Methods section.

We agree with the reviewer. In ITC experiments, samples in the presence of $(GT)_6$ and $(GT)_6A_{12}(GT)_6$ were fitting with the one set of sites model, while samples in the presence of $(GT)_{12}$ were fitting with the two set of sites model. The equations underlying these two fitting models are described in detail in MicroCal PEAQ-ITC Analysis Software user manual. A description and the link of the user manual has been added in the text (Page 69, lines 2444-

2454). The equations used for fitting the SAXS data have been added in the text (Page 71, lines 2529-2536).

Q13: Figures are referenced out of order in some sections (e.g., Fig. S17 etc.).

We agree with the reviewer. We changed the order of Supplementary Figures.

Q14: Some reported fitting errors for the ITC data (Table S2) seem small and should be double-checked.

We have double-checked the values. These values are estimated from the analysis by using Microcal PEAQ-ITC Analysis Software provided by the manufacturer.

Q15: The manuscript would benefit from thorough editing to correct pluralization errors and missing articles throughout the text.

We are sorry for that. We tried to do our best to correct these errors.

Reviewer 3:

Q1: Please deposit (at least WT SAXS data) in the SASBDB.

We thank to the reviewer. The SAXS data has been deposited on the SAXBDB database with the accession code SASDXL8.

Q2: In General consider showing $p(r)$ functions when discussing the changes in size. Kratky plots are shown throughout the manuscript but the information gained from the plot is not stated.

We agree with the reviewer. $P(r)$ function plots for all the samples have been presented in Supplementary Figures 13, 14 and 19.

Q3: In general, the SAXS results could be strengthened by including quantitative comparisons—for example, reporting the D_{max} : for wild-type TDP-43 (162 nm) alongside the range observed for the mutants (159–165 nm).  and quantify accordingly in line 286 as well the increase in D_{max} values (line 297) and later in text. Is there a way to estimate the errors from D_{max} ? (Obviously, the information is given in various tables, but could help the reader).

We thank to the reviewer. The program that we used to estimate the D_{max} is ATSAS-GNOM (shown in SAXS section in Materials and Methods). There is no error estimated for D_{max} that ATSAS could calculate. D_{max} values have been added in the text (Page 7, lines 278-279 and Page 8, lines 306-309).

Q4: Elution profiles

1. Supplementary Figure S9C -- Please comment on the shoulders - such as for the E17R mutants

We thank to the reviewer. The shoulder corresponds to an excess of free proteins. Indeed, as shown in the Supplementary File 1, an excess of protein than oligonucleotide target was used. Moreover, the absorbance value at 280 nm is higher than that at 260 nm (data not shown). Sentences have been added to precise shoulders in the legend of Supplementary Figure 15 (Page 49, lines 1750 and 1754).

Supplementary Figure 11 -- Comment on the shoulders at frame 100 -- are these aggregates - higher oligomer structures?

We thank to the reviewer. The shoulders at frame 100 indeed represents higher oligomeric structures and not aggregates which eluate in the void volume of the column as shown for the protein alone in Supplementary Figure 1. A sentence has been added to precise this point in the

legend of Supplementary Figure 18 (Page 55, lines 1973-1974). However, the signal has been decomposed using the REGALS algorithm available in RAW as explained in the materials and methods part (Page 71, lines 2519-2521).

Q5: line 265---The chromatograms and the SAXS images were recorded during the elution. We observed that the elution volumes were similar for all the conditions in the presence of (GT)₁₂ for which the binding is cooperative." To improve clarity, consider explaining the underlying observations in a bit more detail. E.g. The single mutants (blue and yellow trace) elute similarly as WT (red trace), which shows cooperative binding (see ITC Fig 1D, red plot)

We thank to the reviewer. This following sentence was added "In the presence of (GT)₁₂ for which the binding is cooperative (Figure 1D), we observed that the single mutants (blue and yellow trace) display similar elution volumes as WT (red trace) (supplementary Figure 13A)." (Page 7, lines 267-269).

Q6: Supplementary Figure S9A -- I'm not sure what the difference is between the upper and lower panels. Perhaps a copy/paste issue with the legend?

The upper panel displays absorbance measured at 280 nm, while the lower panel displays absorbance measured at 260 nm. It is now better indicated in the text (Page 44, lines 1591-1592).

Q7: line 260 -- Restructure: To investigate structural changes on a larger scale than accessible by NMR, we performed SAXS analysis of in vitro-purified TDP-43-nucleic acid complexes.

We thank to the reviewer. The sentence has been restructured (Page 7, lines 261-262)

Q8: line 264 -- The acquisition of the SAXS images WAS..... or SAXS measurements were performed in-line with size-exclusion chromatography (SEC)

We thank to the reviewer. This mistake has been corrected (Page 7, line 265).

Q9: line 265 -- specify which chromatogram (UV 280nm + 260nm ?)

In SAXS experiments, we used a combination of SAXS with on-line Size-Exclusion Chromatography (SEC). The SAXS measurement is in-line after the UV measurement. The UV modules can measure simultaneously the absorbance at 260 and 280 nm. SAXS images are recorded just after the UV modules at a frames rate of 1 second per images. We precise SEC chromatogram in the text (Page 7, line 266).

Q10: line 273 -- Already here, place the reference to Figure 3 and SAXS Table in supplementary information. (see line 277)

We agree with the reviewer. This point has been added in the text (Page 7, lines 274-275).

Q11: line 274 -- As two hypotheses were formulated earlier be specific which one is in agreement.

We agree with the reviewer that it was not clear. The sentence has been changed clarify this point (Page 7, lines 275-277).

Q12: Line 1940 -- in Methods the use of REGALS for decomposing overlapping peaks is mentioned -- was this observed in the measurements?

The program REGALS was used only when we have the contamination of higher oligomeric structures. The program REGALS was used to eliminate the impact of higher-order oligomer contamination on the sample's elution peak. Sentences have been added to explain this point in the text (Page 71, lines 2519-2521 and Page 55, lines 1973-1974).

Q13: Line 1945 -- not clear what is meant with: were acquired from averaged SAXS experimental curves.

We calculate an average of several SAXS curves with a similar and stable R_g values to generate a final SAXS curve allowing a better signal/noise ratio. A sentence has been added to clarify this point (Page 71, line 2526-2527).

Q14: Line 1950 -- models ... HAVE been generated

We thank to the reviewer. This mistake is corrected (Page 71, line 2538).

General:

- To improve clarity and conciseness, consider introducing the E52R and R17E mutants, along with the different nucleotide constructs, such as (GT)₆A₁₂(GT)₆, earlier in the text. This would reduce the need for repeated reintroductions later in the manuscript.

We agree. We have limited the number of reintroductions of E17R and R52E in the manuscript.

- line 30 -- is it assembly of TDP-43 in (OR ON ?) introns

We agree. It has been changed into “on introns” (Page 1, line 31).

- line 144 -- better: .. and includes the full NTD domain. (instead of while keeping)

We agree. The sentence has been changed into “including the full NTD domain” (Page 4, lines 143-144).

- line 148 -- ..we compared the chemical shifts of the RRM1-2 residues to constucts containing (1-277) and lacking (101-277) the NTD domain

We agree. The sentence has been changed into “when the fragment contains or lacks the NTD domain” (Page 4, line 148).

- line 167 -- replace whatever with whether

We agree. It has been corrected (Page 4, line 165).

- line 170 -- In THE presence of...

Thanks. “the” has been added in this sentence (Page 4, line 168).

- line 172 -- remove 'a' to read: a major feature of cooperative association with (GT)₁₂

We agree. “a” has been removed (Page 4, line 170).

- line 177 -- perhaps add 'red plot (Figure 1D)

We agree. “(red plot in Figure 1D)” has been added in the text (Page 5, lines 175-176).

- line 178 -- why not mention the second mutant?

We agree. The second mutant E17R has been mentioned in the text (Page 5, line 179).

- line 183 -- add "green plot"

We agree. "(green plot in Figure 1D)" has been added in the text (Page 5, lines 182-183).

- line 214 -- 'A' similar result...

We agree. "The" has been replaced with "A" (Page 5, line 215).

- line 267 -- add: the binding is cooperative (WT AND SINGLE MUTANTS).

The sentence has been changed into "In the presence of (GT)₁₂ for which the binding is cooperative (Figure 1D), we observed that single mutants (blue and yellow trace) display similar elution volumes as WT (red trace) (Supplementary Figure 13A)." (Page 7, lines 267-269).

- line 290 -- remove 'The' before 'similar results'

We agree. "The" has been removed (Page 8, line 309).

- line 317 -- replace 'To document' with to assess or investigate or characterise...

We agree. This point has been replaced (Page 8, line 336).

- line 553 -- mutated TDP/43*aa. 1/277

We agree. "mutated" has been added (Page 15, line 601).

- line 382 -- define 'HA'

We have defined HA-tag in the text (Page 74, lines 2619-2621)

- line 401 -- use passive form : Two pairs of lysine residues, known to undergo acetylation following arsenite stress, were mutated to arginine.

We thank to the reviewer. The sentence has been changed (Page 10, lines 429-430).

- Supplementary Figure 2B: One could first show R52E mutant then the double mutant

We agree. The order has been changed in Supplementary Figure 4B.

- Supplementary Figure 3: Color names should be in lowercase rather than uppercase.

Sorry. The mistake has been corrected in the legend of Supplementary Figure 6 (Page 35, lines 1276-1277).

- line 1089 -- interactions DO not interfere

Sorry. The mistake has been corrected (Page 34, line 1271).

- line 1098 -- Significant CSPs ARE OBSERVED only in the binding residues of RRM1-2 (assigned in black).

We agree. The description has been changed (Page 35, line 1280).

- line 1099 -- COMPARED to

We agree. The description has been changed (Page 35, line 1281).

- line 1142 -- Zoomed-in VIEW on THE chemical shift perturbations FOR TDP-43 residues in the RRM, with or without the indicated nucleotides

We agree. The description has been changed into “Zoomed-in view on the chemical shift perturbations for TDP-43 residues in RRMs in the presence or absence of indicated oligonucleotides” (Page 36, lines 1327-1328).

- line 1144 -- R52E not R525E

Thanks. The mistake has been corrected (Page 36, line 1329).

- line 1927 -- Data collection parameters ARE shown

Thanks. The mistake has been corrected (Page 71, line 2504).

- line 1934 -- In each experiment -- (remove 's')

Thanks. The mistake has been corrected (Page 71, line 2512).

Reviewer #4 (Remarks to the Author):

Thank you for co-reviewing the manuscript.

Revision of manuscript NCOMMS-25-35416B

“From TDP-43/RNA complex formation to disease-linked TDP-43 aggregation through a structural and cellular approach” by Yitian Feng et al.

We would like to thank the reviewers for their comments and constructive suggestions, which have significantly enhanced the quality and clarity of this manuscript.

Point-by-point response to reviewers' comments

Reviewers' comments and our answer (in blue):

REVIEWERS' COMMENTS

Remaining comments to address the previous concerns of reviewer #1:

- Regarding reviewer #1's concern regarding imaging parameters used, please clarify whether there were differences in imaging parameters across experiments and why. This information could be added to the results/methods.

There were no changes in imaging parameters when the same output was produced. Our pipelines are entirely automated. This comment is added to the materials and methods (See page 19).

- Please comment on reviewer #1's comments regarding the interchangeable use of 'granule' versus 'aggregate'. Please define this and update the main text.

We now indicated only “aggregates”. Granules is only mentioned for stress granules, which are reversible. A comment about the nature of the aggregates is indicated in page 10.

- Regarding the rationale for not including data on the 6-point mutant and instead providing SEC data comparing wt and mutant protein. The data does not show complete prevention of interactions, but rather there's at least a reduction. A trend for reduction is shown, but not proof of complete prevention. We kindly request these points are clarified, and the limitations of these data are discussed in the text.

We show in supplementary figure 1 that a single mutant is largely enough to strongly limit NTD multimerization by R52 mutation. First, SEC data combined to SAXS analysis which show clearly that the size of R52E as well as E17R display a MW expected for a monomer form (Supplementary Figure 1). Second, from NMR data, the peaks of interface residues are displayed for R52E and E17R mutants contrary to WT form. Surprisingly, when we incubate R52E and E17R, we could reconstitute a WT behaviour with the disappearance of NMR peaks of interface residues (Supplementary Figure 9). Accordingly, we think that it is useless to perform 6-point mutations along the NTD sequence covering several residues of interface (E14A, E17A, E21A, Q34A, R52A, and R55A) and consequently masks insights into the behaviour and dynamics of numerous

interface residues. Besides potential alterations in NTD structure, the presence of multiple mutations precludes us from carrying out dimer reconstitution using complementary mutations (Figure 2C, Figure 3B, Supplementary Figure 3, 6, 7, 9, 14). A comment was added (page 4).

Reviewer #2 (Remarks to the Author):

I appreciate the authors efforts and their responsiveness to my concerns. I believe all major concerns have been addressed and the manuscript is much improved.

One small remaining concern, I had asked that the authors include the equations they used to fit their data, and while they did this with the SAXS analysis, they provided a link to the ITC instrument's manual. I note that this document is on a password-protected site requiring registration to access. Further, since this is a commercial site, the links and document may be updated/revised/moved/discontinued etc thus not preserving critical information needed to reproduce the results of this paper. For full transparency and completeness, the authors should include the equations used to analyze the ITC data in the material and methods section. If length is a concern, then can be included in an expanded M&M section or supplemental material.

We agree with the reviewer. The equations used for fitting the ITC data have been added in the text (Pages 15-16, lines 604-620).

Reviewer #3 (Remarks to the Author):

All my comments have been addressed, and several figures have been added. In addition, some text passages have been extended for greater clarity. I recommend that the manuscript be accepted for publication.